# A fitness trade-off between seasons causes multigenerational cycles in phenotype and population size

**Gustavo S Betini\*, Andrew G McAdam, Cortland K Griswold, D Ryan Norris**

Department of Integrative Biology, University of Guelph, Guelph, Canada

**Abstract** Although seasonality is widespread and can cause fluctuations in the intensity and direction of natural selection, we have little information about the consequences of seasonal fitness trade-offs for population dynamics. Here we exposed populations of *Drosophila melanogaster* to repeated seasonal changes in resources across 58 generations and used experimental and mathematical approaches to investigate how viability selection on body size in the non-breeding season could affect demography. We show that opposing seasonal episodes of natural selection on body size interacted with both direct and delayed density dependence to cause populations to undergo predictable multigenerational density cycles. Our results provide evidence that seasonality can set the conditions for life-history trade-offs and density dependence, which can, in turn, interact to cause multigenerational population cycles.

**\*For correspondence:** gsbetini@gmail.com

**Competing interests:** The authors declare that no competing interests exist.

## Introduction

In many organisms, reproduction is confined to seasonal fluctuation in periods of high resource, in which both fecundity (reproduction) and viability (survival) selection can occur, and periods of low resources, when reproduction stops and natural selection occurs only through viability selection. Consequently, sequential episodes of reproduction and survival caused by seasonality could be a major source of fluctuations in the strength and direction of natural selection (*Darwin, 1859*; *Lack, 1954*; *Fretwell, 1972*; *Schluter et al., 1991*; *Bell, 2010*; *Bergland et al., 2014*), giving rise to classic life-history trade-offs (*Lack, 1947*; *Roff, 1992*; *Stearns, 1992*; *Garland, 2014*). More specifically, traits that confer a fecundity advantage, but which are associated with a survival cost, will experience natural selection in one season that is opposed by selection in the subsequent season (*Levins, 1968*; *Michod, 2006*; *Bell, 2010*; *Bergland et al., 2014*). The sequential rather than simultaneous nature of trade-offs driven by seasonality could have important consequence for the trait distribution within and across generations (*Levins, 1968*; *Grafen, 1988*; *Michod, 2006*) and population dynamics (*Ozgul et al., 2010*).

One way by which life history trade-offs might arise from seasonal variation in resources is via body size (*Ozgul et al., 2010*, *2014*). Large individuals usually have higher fecundity (*Mueller and Joshi, 2000*; *Schulte-Hostedde and Millar, 2004*), but they could also have lower survival during the non-breeding season when resources are scarce (*Stockhoff, 1991*; *Reznick et al., 2000*; *Munch et al., 2003*; *Monaghan, 2008*). In addition, large individuals might take more time to grow and require more resources for maintenance (*Munch et al., 2003*), which could negatively impact their survival probability (*Kingsolver and Huey, 2008*).

This association between fitness and body size in seasonal environments could have important consequences for population dynamics, particularly when selection on body size is density-dependent (*Mueller, 1997*; *Sinervo et al., 2000*; *Travis et al., 2013*). Differences in the selective advantage of body size across seasons could also shed light on the long-standing question about why

**eLife digest** Many wild populations go through long cycles in abundance that span several generations. The traditional explanation for such "multigenerational" cycles is that they are driven by predator/prey relationships, the classic example being oscillations between the numbers of lynx and snowshoe hares.

Population cycles could also be driven by seasonal changes. For example, traits that help animals to produce large numbers of offspring during the breeding season may reduce the ability of the animal to survive the non-breeding season. Body size is one such trait. Large individuals tend to produce more offspring, but their larger body size means that they find it harder to survive when food is scarce. As a consequence, large individuals should have an advantage and be more common when the population size is low and there are enough resources for all individuals. However, small individuals should be more abundant when population size is high. This trade-off caused by seasonality could set the population in motion towards predictable, multigenerational cycles.

To test this idea, Betini et al. established populations of fruit flies that went through 'breeding' and 'non-breeding' seasons. This was achieved by periodically altering the flies' food to prevent the females from laying eggs (in the lab, fruit flies do not normally have non-breeding seasons). Over 58 generations, the number of flies in each population cycled between peaks of high and low numbers.

When the population contained relatively few flies, there was strong selection for large flies because they have high reproductive success. Hence, the population grew. When the population was large, meaning that the flies had to compete for a limited amount of food, there was strong selection for small flies because they are better able to survive on limited resources. However, small flies also produce fewer offspring on average, resulting in a decrease in population size. When the flies all had sufficient food during the non-breeding season, these regular cycles completely disappeared.

A major challenge will be to understand how common this phenomenon is in the wild. Virtually all organisms live in seasonal environments but whether they face strong trade-offs in the expression of traits is not well understood. This is primarily because of the difficulty in following individuals throughout the year.

population densities of many species fluctuate periodically over time (*Elton and Nicholson, 1942*; *Kendall et al., 1999*; *McCauley et al., 2008*; *Yan et al., 2013*). For example, when body size is positively related to fecundity, but small individuals survive better in the non-breeding season (*Stockhoff, 1991*; *Munch et al., 2003*; *Monaghan, 2008*; *Betini et al., 2014*), these opposing patterns of selection could cause population cycles if selection is density-dependent. Specifically, if smaller offspring have higher survival when density is high, then the population will be composed of smaller than average individuals with lower average fecundity in the following breeding season. This lower mean fecundity will reduce population growth rates even though larger individuals will be favoured through fecundity selection. As population size declines, the strength of density-dependent viability selection on body size will also decline, which could cause net selection to reverse and favour larger individuals due to the fecundity benefit of being larger. As large individuals increase in frequency, population size should also increase via an improvement of reproductive output, returning populations to high densities. Although changes in the intensity and direction of natural selection caused by environmental variation are widespread (*Schluter et al., 1991*; *Bell, 2010*; *Thompson, 2013*; *Bergland et al., 2014*), we have little information about whether opposing episodes of natural selection could arise from seasonality and indirectly affect population dynamics through the feedback loop between ecological (density dependence) and evolutionary (selection and evolution) processes (*Chitty, 1960*; *Krebs, 1978*; *Hairston et al., 2005*; *Pelletier et al., 2009*; *Ozgul et al., 2010*; *Schoener, 2011*).

Here, we investigated how a seasonal fitness trade-off related to body-size could affect changes in population size and body size over time using replicate populations of *Drosophila melanogaster* exposed to repeated changes in food resources. In addition to standard breeding conditions for *Drosophila*, we also created a 'non-breeding season' by manipulating the food medium to prevent

females from laying eggs during this period. Thus, in this system, breeding and survival were restricted to two sequential and distinct seasons (hereafter 'breeding' and 'non-breeding'). The number of days and amount of food in each season was determined so that both fecundity and non-breeding survival were density-dependent (*Betini et al., 2013a*, *2014*, *2015*), which is an important feature of many populations. In *Drosophila*, as in many other species, the positive correlation between body size and fecundity is well known (*Mueller and Joshi, 2000*; Appendix 1) and we previously demonstrated that small individuals have higher survival during the non-breeding season when abundance is high (*Betini et al. 2013a*, *2014*). In addition, populations of *D. melanogaster* do not show evidence of multigenerational cycles (*Mueller and Joshi, 2000*), even when kept under the same conditions as our breeding season (i.e. 'aseasonal populations'; Appendix 2). We, therefore, hypothesized that density dependence and opposing episodes of fecundity and viability selection on body resulting from seasonality could cause predictable and repeatable fluctuations in both population and body size. Specifically, we predicted that seasonal fitness trade-offs would cause population size and body size to undergo multigenerational cycles between periods of high abundance, when small individuals predominate, and periods of low abundance, when large individuals are more frequent. Furthermore, we predicted that populations not exposed to viability selection in the non-breeding season would lack periodic fluctuations in population size and body size.

We tested whether seasonality could result in multigenerational cycles in population size and body size using three experiments (*Figure 1*). In the first experiment, we submitted 45 replicate populations to the seasonal treatment described above and tracked the total number of individuals and body size at the end of the breeding and non-breeding season for 58 generations (the 'long-term control' treatment; *Figure 1*). In the second experiment, we tested the role of viability selection during the non-breeding season by tracking 13 additional populations over 31 generations using a similar protocol to the 'long-term control', but in which we experimentally prevented viability selection in the 'non-breeding' season by providing high levels of food during this season (the 'stop-selection' treatment; *Figure 1*). This protocol also maintained direct density effects on fecundity and

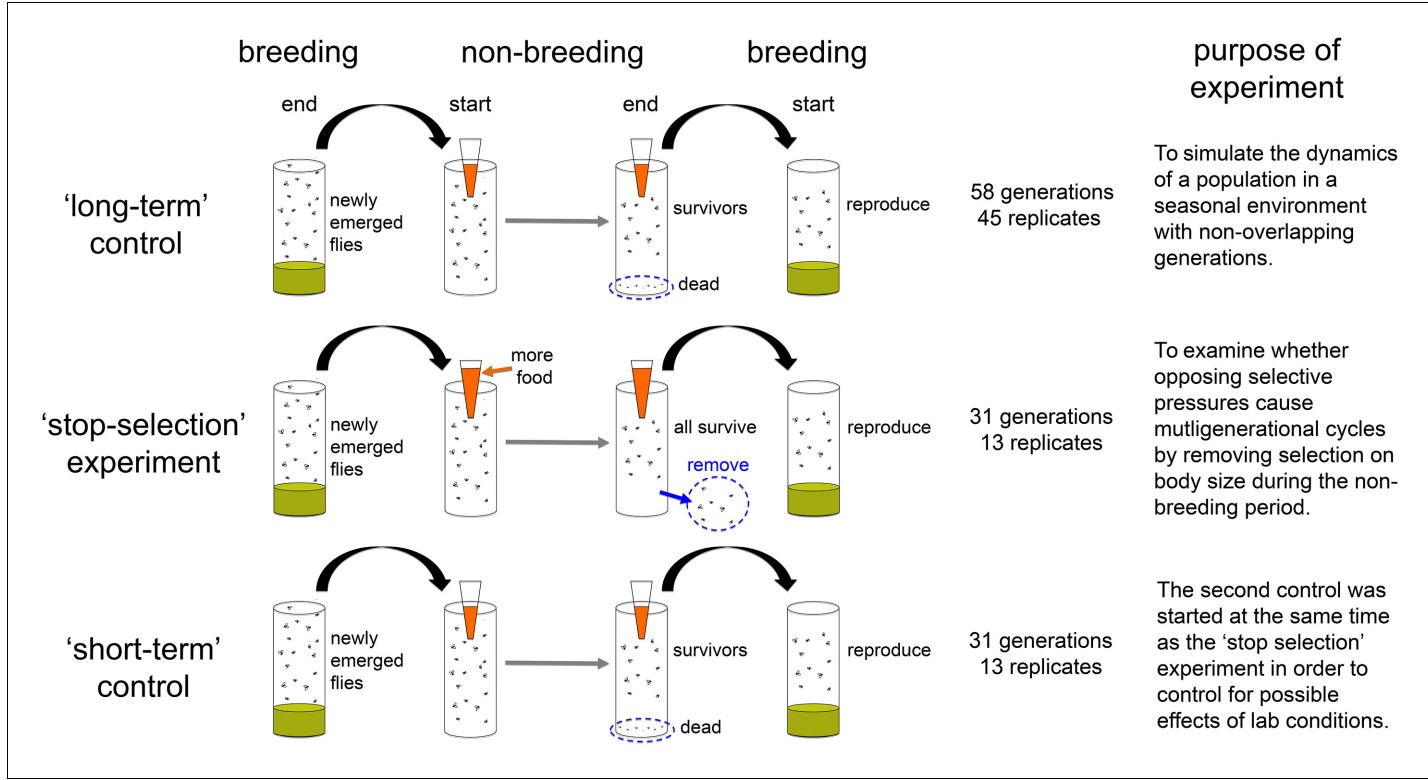

**Figure 1.** A schematic of the three experiments conducted in this study with accompanying duration, number of replicates and brief summary of their purpose.

survival, similar to the ones observed in the 'long-term control'. In order to address potential environmental changes in the lab, we conducted a third experiment using the same protocol as in the 'long-term control', but under the same initial conditions and at the same time as the 'stop selection' treatment. This 'short-term control' experiment also had 13 replicate populations tracked over 31 generations (*Figure 1*).

In addition to these experiments, we also developed a mathematical model to investigate the contributions of both viability selection and delayed density dependence to population dynamics. The 'stop-selection' experiment was designed to eliminate viability selection, but might have also reduced potential effects of past densities on fecundity and survival. Such delayed density-dependent effects can also cause populations to cycle (*Stenseth et al., 2003*; *Yan et al., 2013*), or lead to more complex dynamics, such as chaos (May 1973). One potential mechanism for delayed density dependence are carry-over effects, which we have previously identified in this seasonal system (*Betini et al., 2013a*). Thus, we first investigated if lag effects were present in all three experiments, and then used the mathematical model to understand whether they played a role in the dynamics of their populations.

## Results

### Long-term control

Over 58 generations, the 45 replicated seasonal populations showed a predictable increase in abundance during the breeding season, where the food medium allowed females to lay eggs, and a decline in the subsequent non-breeding season (*Figure 2a*), as is typical of many natural seasonal systems. However, the autocorrelation functions (ACF) also revealed that these short, seasonal cycles were embedded within longer multigenerational cycles where average population size fluctuated 3-fold (insert in *Figure 2a*). In these populations, the ACF function was characterized by stationary periodic dynamics, which resulted in an oscillatory decay to zero (*Figure 2a* inset).

To investigate the presence of viability selection for small body size and whether this selection was density-dependent, we measured female dry weight in 38 generations from 25 different populations. As expected, there is a negative correlation between population size at the end of the non-breeding season and body size at the end of the non-breeding season (Pearson's product-moment correlation = $-0.64$; $t = -4.48$, p<0.001), suggesting that density negatively impact body size in the non-breeding season. Mean survival during the non-breeding season was 71% ($\pm 0.21$ SD) and survival was density-dependent ($\beta_{survival} = -0.001$, t = $-27.73$, p<0.001). Mean female dry weight was significantly lower after the non-breeding season (0.279 mg, $n = 3620$ females) than before the non-breeding season (0.381 mg; $n = 5258$ females; standardized values = 0.577 before and $-0.566$ after the non-breeding season; Welch t-test: t = $-35.90$, df = 1,589.240.24, p<0.001; *Figure 2b*) and this viability selection was density-dependent (*Figure 2c*; *Table 1*, Appendix 3). That is, when population size was high at the start of the non-breeding season, there was stronger selection for smaller flies and this was driven by changes in mean dry weight after the non-breeding season rather than changes in the mean dry weight before the non-breeding season (Appendix 3). Average dry weight measured after the non-breeding season also showed multigenerational cycles (*Figure 2d*), as indicated by the autocorrelation function (*Figure 2d*, inset), varying between average peaks of 0.32 mg and lows of 0.23 mg.

### 'Stop selection' experiment

In order to test the role of viability selection in these multigenerational cycles, we experimentally eliminated viability selection during the non-breeding season in 13 additional populations. Unlike the 'long-term controls' (*Figure 2a*), there was no evidence of multigenerational population cycles in these 'stop selection' populations (*Figure 3a* inset). In addition, body size did not significantly decline after the non-breeding season (*Figure 3b*; average body size was 0.337 mg before and 0.331 after the non-breeding season; $n = 1290$ and $n = 689$ females, respectively; standardized values: 0.028 before and $-0.052$, respectively; Welch t-test: $t = -1.733$, df = 1,479.400.40, p=0.081). There was also no evidence of density-dependent selection (*Figure 3c*; *Table 1*) and no evidence of cycles in body size (*Figure 3d* inset).

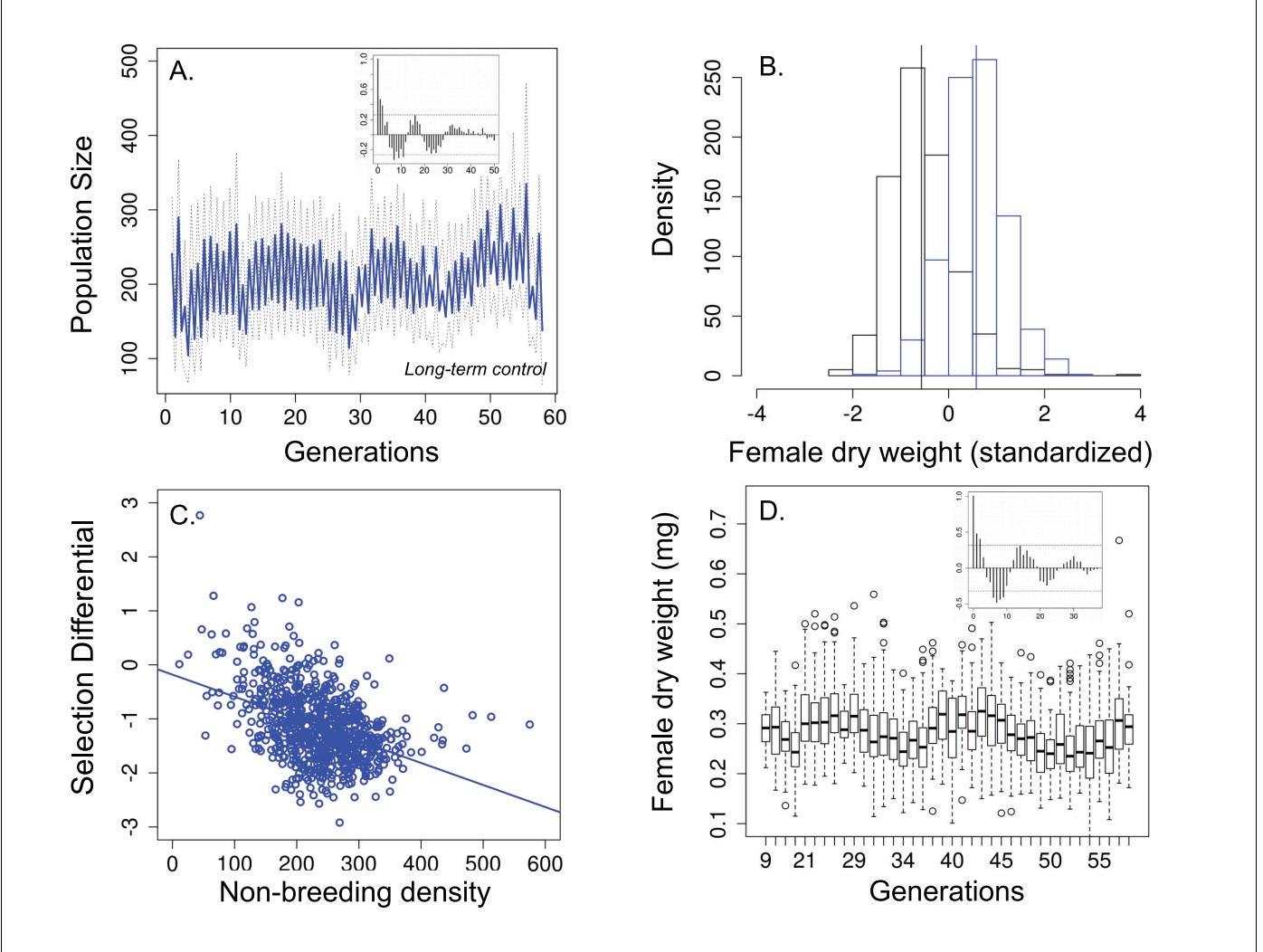

**Figure 2.** Population size, changes in body size and selection differentials for body size in the 'long-term control' experiment. (a) Population size of seasonal flies cycled over 58 generations; (b) Female dry weight before (blue bars) and after (black bars) the non-breeding season. Vertical bars indicate the mean female dry weight before and after the non-breeding season; (c) Increased population size in the non-breeding season led to stronger directional selection for smaller flies. (d) Time series of female dry weight measured at the end of the non-breeding season over 38 generations. In (a and (d), the autocorrelograms (insets) showed evidence of cycles in both population size and body size. In (a solid blue line denotes mean population size for each generation from all replicates and dotted lines denote ±1 s.d. In (d, the horizontal line within each box represents the median value, the edges are 25th and 75th percentiles, the whiskers extend to the most extreme data points, and points are potential outliers.

## 'Short-term control'

Over 31 generations, and similar to the 'long-term control' (*Figure 2a*), the 'short-term control' exhibited evidence for multigenerational cycles (*Figure 4a*). These 'short-term control' populations also experienced viability selection that was density-dependent. Overall, female dry weight significantly decreased from an average of 0.371 mg before the non-breeding season (n = 968 females) to 0.276 mg after the non-breeding season (n = 623 females; standardized values = 0.487 before and −0.761 after the non-breeding season; Welch t-test: t = −30.601, p<0.001; *Figure 4b*), but the magnitude of this viability selection was stronger when densities were higher at the start of the non-breeding season (*Table 1*, *Figure 4c*).

**Table 1.** Parameter estimates obtained from linear mixed effect models to investigate viability selection on body size as a function of thenumber of individuals at the beginning of the non-breeding season. In the 'long-term control', $R^2_{LMM(m)}$=0.18 and $R^2_{LMM(c)}$=0.20; in the 'stop selection' treatment, $R^2_{LMM(m)}$=0.006 and $R^2_{LMM(c)}$=0.006; and in the 'short-term control', $R^2_{LMM(m)}$=0.22 and $R^2_{LMM(c)}$=0.22. $R^2_{LMM(m)}$ is the variance on the response variable that is explained only by the fixed effects and $R^2_{LMM(c)}$ is the variance that is explained by both fixed and random effects. In all models, the selection differential was the response variable, abundance at the beginning of the non-breeding season was the fixed effect and population (vial) was the random effect.

| Parameters | Fixed effects estimate | SE | Df | T | P |
|---|---|---|---|---|---|
| **1. Long-term control** | | | | | |
| Intercept | −0.181 | 0.081 | 434.5 | −2.22 | 0.027 |
| Non-breeding abundance | −0.004 | 0.003 | 745.3 | −12.67 | <0.001 |
| **2. Stop selection** | | | | | |
| Intercept | −0.341 | 0.159 | 159 | −2.15 | 0.033 |
| Non-breeding abundance | 0.001 | 0.001 | 159 | 1.06 | 0.291 |
| **3. Short-term control** | | | | | |
| Intercept | 0.028 | 0.184 | 100 | −0.15 | 0.880 |
| Non-breeding abundance | −0.005 | 0.001 | 139 | −6.44 | <0.001 |

## Delayed density dependence

We statistically investigated whether fecundity and survival were influenced by density in past seasons (i.e. delayed density dependence) in all three experiments: 'long-term control', 'short-term control' and 'stop-selection'. We used linear mixed effect models with vial (population) as a random effect and densities going back up to two generations as fixed effects. In the 'long-term control' and 'short-term control', fecundity and survival were influenced by density in the current and past season (*Table 2* and *Table 3*). In contrast, the 'stop-selection' experiment showed evidence of the delayed effects on fecundity (*Table 2*), but not on survival (*Table 3*), meaning that the 'stop-selection' treatment eliminated both viability selection on body size as well as delayed effects of density on survival.

## Mathematical model

A mathematical model including delayed density dependence as well as the effects of body size on survival (viability selection) and fecundity resulted in multigenerational cycles in population size (before red line in *Figure 5a*), similar to those observed in our 'long-term control' and 'short-term control' (*Figure 2a* and *Figure 4a*, respectively). The model without viability selection on body size and delayed density dependence (i.e. including only effects of current abundance on fecundity and survival), resulted in the elimination of multigenerational cycles (after red line in *Figure 5a*), as observed in our 'stop selection' populations (*Figure 3a*). The exclusion of only viability selection eliminated the fitness trade-off in body size and allowed larger flies to survive to breed. This led to unstable population dynamics (i.e. the population crashed after 10 generations; *Figure 5b*) because larger flies have greater fecundity and there was a negative interaction between body size and abundance. The exclusion of delayed density dependence alone eliminated the cycles (*Figure 5c*), suggesting that viability selection and delayed density dependence are necessary for these persistent multigenerational cycles to occur. The model with low heritability ($h^2 = 10^{-5}$) also generated multigenerational population cycles (*Figure 5d*).

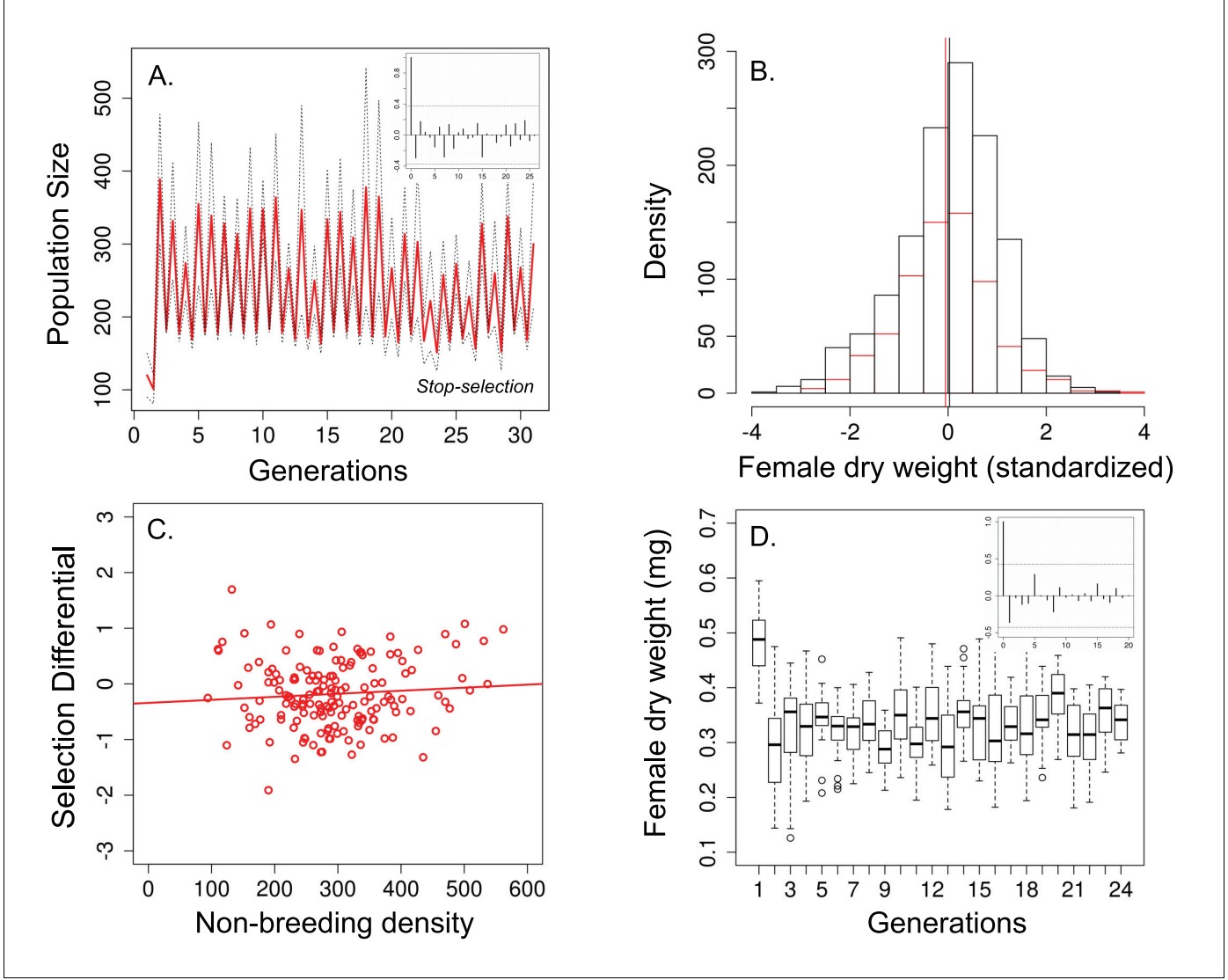

**Figure 3.** Population size, changes in body size and selection differential for body size in the 'stop-selection' experiment. (a) Population size of seasonal flies cycled over 31 generations. Unlike the 'long-term control, (b) there was no significant change in body size after the non-breeding season (female dry weight before - red bars - and after -black bars- the non-breeding season; vertical bars indicates the mean female dry weight before and after the non-breeding season) and (c) no evidence that selection for smaller flies was density dependence. (d) Time series of female dry weight measured at the end of the non-breeding season over 27 generations. In (a) and (d), the autocorrelograms (insets) showed no evidence of cycles in population or body size. In a solid red line denotes the mean population size for each generation from all replicates and dotted lines denote ±1 s.d. In (d), the horizontal line within each box represents the median value, the edges are 25th and 75th percentiles, the whiskers extend to the most extreme data points, and points are potential outliers.

## Discussion

Our results provide empirical and mathematical evidence that the interplay between density-dependence and evolutionary trade-offs caused by seasonality can have important consequences for population dynamics. In our experimental system, seasonal variation in resources resulted in a fitness trade-off and multigenerational population cycles. These cycles were observed in the absence of predation, long-term fluctuations in resources, or any explicit negative-frequency dependence, all of which are known to cause regular fluctuations in population size (*Lindström et al., 2001*; *Yan et al., 2013*). Thus, the emergence of population cycles caused by three common characteristics of natural populations (seasonality, a life-history trade-off and density-dependence), suggests that these

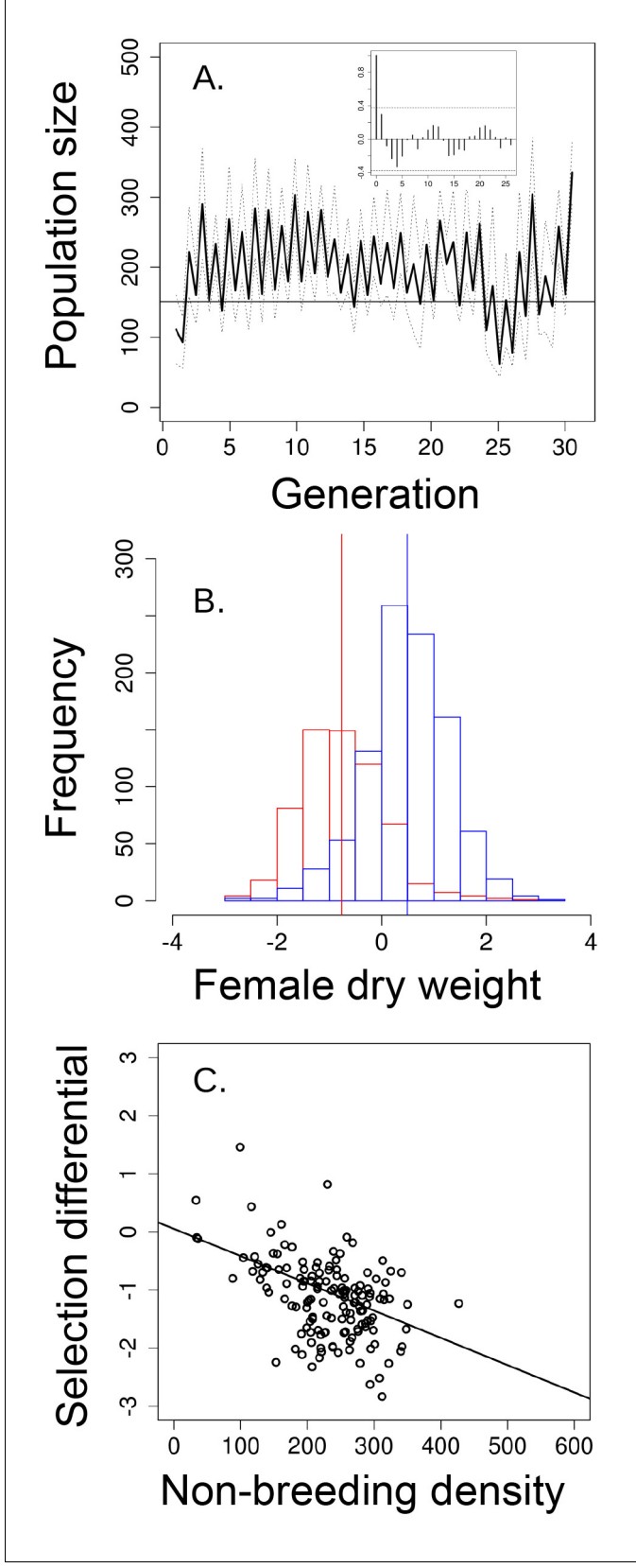

**Figure 4.** Population size, changes in body size and selection differentials for body size in the 'short-term control' experiment. (a) Population size of seasonal flies cycled over 31 generations, as suggested by the autocorrelogram
*Figure 4 continued on next page*

*Figure 4 continued*

(insets), (b) female dry weight before the non-breeding season (blue bars) was higher than after the non-breeding season (red bars; vertical bars indicate the mean female dry weight before and after the non-breeding season). (c) increased population size in the non-breeding season led to stronger directional selection for smaller flies. In (a) solid black line denotes mean population size for each generation from all replicates and dotted lines denote ±1 s.d. In (d, the horizontal line within each box represents the median value, the edges are 25th and 75th percentiles, the whiskers extend to the most extreme data points, and points are potential outliers.

ecological and evolutionary processes could contribute to fluctuations in population size over a wider range of taxa and environments than has previously been considered. Our experimental elimination of viability selection and delayed density-dependence and our mathematical model confirmed the importance of both of these evolutionary and ecological processes in the persistence of multi-generational population cycles resulting from seasonal environments.

Recently, it has been proposed that eco-evolutionary dynamics are essential for understanding oscillations in population size (*Hairston et al., 2005*; *Hiltunen et al., 2014*), but evidence for evolutionary change (i.e. genetically based change in ecologically relevant traits) to feedback and influence demography and population dynamics is rare (*Schoener, 2011*; *Hendry, 2013*); but see (*Cameron et al., 2013*). Our results suggest that the evolutionary response to selection might not be required to observe feedback between ecological and evolutionary processes, as long as selection is strong. In flies, as in many other species, body size is highly heritable (*Prout and Barker, 1989*), and it is reasonable to suppose that, in our populations, offspring from smaller flies tend to be smaller. However, even when genetic variation is low, strong selection can still affect trait distributions within generations (*Grafen, 1988*), potentially affecting population size. In our experimental

**Table 2.** Parameter estimates obtained from linear mixed effect models to investigate the effects of current and past density on fecundity in the 'long-term control', 'stop-selection' and 'short-term control' experiments. B refers to population size at the beginning of the breeding season in the current season and NB refers to population size at the beginning of the previous non-breeding season. In the 'long-term control', $R^2_{LMM(m)}=0.34$ and $R^2_{LMM(c)}=0.37$; in the 'stop selection' treatment, $R^2_{LMM(m)}=0.07$ and $R^2_{LMM(c)}=0.07$; and in the 'short-term control', $R^2_{LMM(m)}=0.30$ and $R^2_{LMM(c)}=0.30$.

| Parameters | Fixed effects estimate | SE | T | P |
|---|---|---|---|---|
| **1. Long-term control** | | | | |
| Intercept | 0.450 | 0.012 | 37.445 | <0.001 |
| B | −0.212 | 0.008 | −25.82 | <0.001 |
| NB | −0.008 | 0.008 | −1.05 | 0.296 |
| B * NB | 0.026 | 0.005 | 5.07 | <0.001 |
| **2. Stop-selection** | | | | |
| Intercept | 0.551 | 0.024 | 22.91 | <0.001 |
| B | 0.150 | 0.053 | 2.82 | 0.005 |
| NB | −0.156 | 0.034 | −4.53 | <0.001 |
| B * NB | 0.0613 | 0.03 | 2.31 | 0.021 |
| **2. Short-term control** | | | | |
| Intercept | 0.532 | 0.019 | 27.72 | <0.001 |
| B | −0.226 | 0.024 | −9.34 | <0.001 |
| NB | −0.039 | 0.022 | −1.74 | 0.083 |
| B * NB | 0.046 | 0.017 | 2.65 | 0.008 |

**Table 3.** Parameter estimates obtained from linear mixed effect models to investigate the effects of current and past density on survival in the 'long-term control', 'stop-selection' and 'short-term control' experiments. NB refers to population size at the beginning of the non-breeding season in the current generation. B1, NB1, B2 and NB2, refers the population size at the beginning of each season going back 1 or two generations, respectively. In the 'long-term control', $R^2_{LMM(m)}$=0.35 and $R^2_{LMM(c)}$=0.36; in the 'stop selection' treatment, $R^2_{LMM(m)}$=0.99 and $R^2_{LMM(c)}$=0.99; and in the 'short-term control', $R^2_{LMM(m)}$=0.43 and $R^2_{LMM(c)}$=0.43.

| Parameters | Fixed effects estimate | SE | T | P |
|---|---|---|---|---|
| **1. Long-term control** | | | | |
| Intercept | −0.353 | 0.006 | −54.74 | <0.001 |
| NB | −0.186 | 0.006 | −31.19 | <0.001 |
| B1 | 0.106 | 0.008 | 13.76 | <0.001 |
| NB1 | −0.056 | 0.007 | −8.34 | <0.001 |
| B2 | 0.044 | 0.007 | 5.67 | <0.001 |
| NB2 | −0.020 | 0.007 | −2.89 | 0.004 |
| **2. Stop-selection** | | | | |
| Intercept | −0.489 | 0.001 | −1,068.62 | <0.001 |
| NB | −0.213 | 0.001 | −412.71 | <0.001 |
| B1 | 0.001 | 0.001 | 1.325 | 0.186 |
| NB1 | −0.001 | 0.001 | −1.235 | 0.217 |
| B2 | 0.001 | 0.001 | 0.479 | 0.632 |
| NB2 | 0.001 | 0.001 | 0.225 | 0.822 |
| **3. Short-term control** | | | | |
| Intercept | −0.427 | 0.013 | −33.36 | <0.001 |
| NB | −0.223 | 0.0142 | −15.64 | <0.001 |
| B1 | 0.144 | 0.015 | 7.22 | <0.001 |
| NB1 | −0.037 | 0.015 | −2.40 | 0.017 |
| B2 | 0.041 | 0.020 | 2.03 | 0.043 |
| NB2 | −0.015 | 0.016 | −0.933 | 0.351 |

system, selection for body size during the non-breeding season was strong enough to drive observed cycles, potentially in the absence of any genetic change across generations. Simulations with our mathematical model with essentially zero heritability ($h^2 = 10^{-5}$ as opposed to 0.3) also generated multigenerational population cycles because viability selection changed the distribution of body size, which subsequently affected fecundity. Thus, changes in the trait distribution within a generation caused by seasonal fitness trade-offs could generate feedback loops between ecological and evolutionary process in traits across generations.

We showed that variation in non-breeding abundance can cause density-dependent selection, but our mathematical model suggested that long-lasting effects of past density on survival were essential to generate the multigenerational cycles. These effects were strong in both the 'long-term control' and 'short-term control', but not in the 'stop-selection' treatment. One mechanism that can generate delayed effects is a density-mediated carry-over effect. We have shown how competition during the non-breeding season can negatively impact the physiological conditions of the survivors, reducing their breeding output in the following breeding season (*Ratikainen et al., 2008*; *Betini et al., 2013a*). This carry-over effect could also negatively impact survival if individuals during the non-breeding season were in lower body condition because of high abundance at the beginning of the season. Delayed density dependence can also arise from maternal effects, whereby offspring

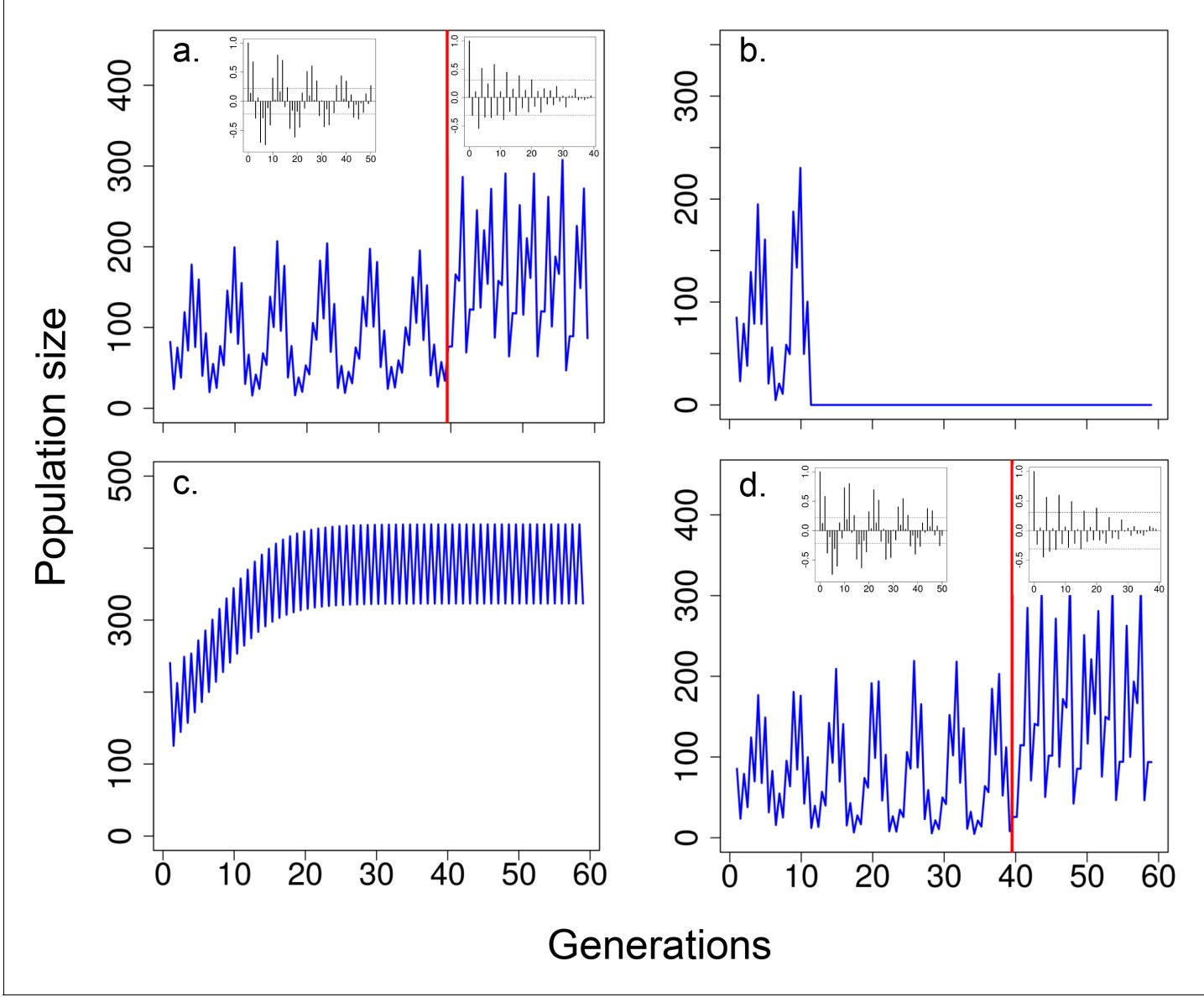

**Figure 5.** Predicted population size according to the integral projection model. Time series generated with the model where, (a) for the first 40 generations (before vertical red line), when both viability selection and delayed density effects on survival during the non-breeding season and fecundity were operating and, for the last 20 generations, only the effects of current population size on fecundity and survival were modelled (i.e. no viability selection and no delayed density effects); (b) population size excluding the effects of viability selection (only the effects of delayed density dependence on fecundity and survival were operating) or (c) excluding the effects of delayed density dependence (only the effects of viability selection were operating). In (d), times series were generated as in (a) but with low heritability (see Results for details). In (b), population crashed (i.e. population size <1) at generation 10. In both (b) and (c), the model incorporated the effects of current abundance on fecundity and survival and heritability for body size. In (a) and (d) inset figures represent autocorrelograms for population size obtained from the mathematical model including delayed density dependence, viability and fecundity selection (before red line) or without both delayed density dependence and viability selection (after red line). See Material and methods for details and model parameters.

attributes are affected by parental abundance, which have previously been shown to generate population cycles (*Plaistow and Benton, 2009*). Both carry-over effects and maternal effects have the potential to interact with viability and fecundity selection because they can affect individual fecundity and survival via changes in body size and condition.

Our mathematical model also suggested that the delayed density dependence alone would cause short cycles and population crashes after only a few generations. It is clear from the model that the addition of viability selection allowed for sustained multigenerational cycles, as was observed in our empirical results. These results are in general agreement with previous studies. For example, in a predator-prey system, evolutionary changes in the prey population extended density cycles compared to the typical quarter-period phase lags between prey and predator densities (*Yoshida et al., 2003*). This evolutionary effect is now believed to be widespread in lab systems (*Becks et al., 2012*; *Hiltunen et al., 2014*). Thus, it is possible that evolutionary processes affecting trait distributions within and between generations can buffer strong negative feedback caused by ecological processes that are typical in many natural systems (e.g. overcompensation caused by density dependence), representing a different form of evolutionary rescue (*Bell, 2013*; *Carlson et al., 2014*).

Opposing viability and fecundity selection on body size was necessary for the persistence of multigenerational population cycles. Such fitness trade-offs between seasons are likely more common than currently appreciated (*Schluter et al., 1991*; *Schmidt et al., 2005*; *Schmidt and Conde, 2006*, *Schmidt and Paaby 2008*; *Kingsolver and Diamond, 2011*; *Behrman et al., 2015*), especially given that many organisms use vastly different habitats over the annual cycle. Although body size caused fitness trade-offs between seasons in our experimental system, many other traits might also exhibit seasonal trade-offs. For example, sexually selected traits enhance mating success, but might also reduce survival during the non-breeding season (*Emlen, 2001*). Alternatively, increased development rates can accelerate age of first reproduction, but are often associated with reduced survival during the adult stage (*Stearns, 1992*; *Kingsolver and Huey, 2008*).

Seasonal fitness trade-offs also represent a form of balancing selection, which can maintain increased levels of genetic variation at evolutionary equilibrium (*Barton and Keightley, 2002*). Seasonal fluctuations in selection might allow for the maintenance of increased adaptive potential in populations (*Shaw and Shaw, 2014*), facilitating more rapid responses to directional environmental change (*Bradshaw and Holzapfel, 2006*; *Bell, 2010*; *Huang et al., 2016*). This temporal variation in selective pressures might also help to explain why some species are more successful at colonizing new areas, as suggested by studies of seasonal changes in genetic variation in wild populations of *D. melanogaster* (*Schmidt and Conde, 2006*; *Bergland et al., 2014*; *Behrman et al., 2015*). Thus, the consequences of seasonal fluctuations in resources are, therefore, not restricted to purely ecological pathways. Seasonality can also alter the strength and direction of natural selection and might maintain a population's adaptive potential over longer evolutionary timescales.

Examples of delayed density dependence causing cycles have been documented in natural populations subjected to strong seasonality (*Merritt et al., 2001*; *Stenseth et al., 2003*; *Yan et al., 2013*). We showed how opposing episodes of selection driven by a life-history trade-off arise in seasonal environments, which could interact with delayed effects to influence population size. Recently, it has been proposed that results from lab studies on consumer-resource dynamics were likely confounded by fast evolutionary changes (*Hiltunen et al., 2014*). The same could be true for lab and field studies on population cycles, if changes in trait distributions are linked to population size. To understand whether this is a common phenomenon in natural populations, one needs to tease apart the evolutionary from the ecological consequences of variation in density. As in our lab system, this is a constraint that needs to be overcome with either improved controlled experiments or mathematical approaches. Nevertheless, given that variation in density is common in many natural systems, measurements of selection and population density in both seasons will provide a better understanding of the dynamics of natural populations and how environmental change might alter such dynamics.

## Materials and methods

### 'Long-term control' experiment

To simulate seasonality in populations with non-overlapping generations, we changed food composition to generate two distinct 'seasons' (n = 45 populations). During the breeding season, we placed adults in vials (28 × 95 mm) with a dead yeast-sugar medium to lay for 24 hr (day 0), after which adults were discarded and larvae were allowed to mature to adults (16 days). Individuals were then transferred (day 17) to a non-breeding environment for 4 days. The non-breeding season consisted

of an empty vial of the same size as the breeding vials, but food was provided by a pipette tip filled with 0.200 ml of 5% water–sugar solution per day from the top of the vial. This solution was sufficient for many flies to survive (~95% survival at low population size) but did not allow females to lay eggs (*Bownes and Blair, 1986*; *Betini et al., 2013a*). In both seasonal treatments, a consistent amount of food was provided regardless of population size, which caused reproduction and survival to be density-dependent (*Betini et al., 2013a*). Each of 45 replicate populations was repeatedly transferred between breeding and non-breeding environments for 58 generations and together they are referred to as 'long-term controls'. The quality and amount of the medium during the breeding season mimics the scenario well explored in other *Drosophila* systems (*Mueller and Joshi, 2000*), where the food is of lower quality and more limited for adults compared to larvae. Dynamics of populations experiencing only this medium (i.e. only the breeding season) do not show evidence of cycles (*Mueller and Joshi, 2000*); (*Appendix 2—figure 1*).

## 'Stop-selection' experiment

To investigate whether opposing selective pressures caused multigenerational cycles in seasonal populations, we experimentally stopped viability selection for a smaller body size in the non-breeding season while preserving density-dependent survival. To do this, we exposed 13 new replicate populations to the same seasonal change in food resources (described above) over 31 generations, but provided unlimited access to food during the non-breeding season (0.8 ml/day instead of 0.2 ml/day; initial population size was 5 males and 5 females). Average mortality was reduced from 28% (±0.4; mean ± s.e.) in the 'long-term control' populations to 0.5% (±1) in these new 'stop selection' treatments. After four days in the non-breeding season, we haphazardly selected the survivors that moved to the breeding season. To preserve the density-dependent survival process during the non-breeding period, the number of survivors for each population was calculated based on a logistic non-breeding survival function parameterized from our 'long-term control' populations (*Wilson, 1994*; *Figure 6*)

$$Su = \frac{1}{1 + \left(\frac{N}{v}\right)^w}$$

where $Su$ is survival (number of survivors at the end of the non-breeding season divided by the total number of individuals at the beginning of the non-breeding season), $N$ is the population size at the beginning of the non-breeding season, and $v$ and $w$ are constant to be estimated from the data. The function was parameterized with our 'long-term control' populations (45 replicates over 43 generations) using the non-linear function *nls* in R (*R Core Team, 2015*). Individuals that moved to the following breeding season were haphazardly selected.

## 'Short-term control' experiment

Our 'long-term control' and 'stop-selection' populations were initiated at different periods and with different population sizes. Thus, differences in the lab environment and initial conditions could have influenced the dynamics of these populations. For these reasons, we also initiated an additional 13 replicate populations at the same time and same initial population size as the 'stop-selection' populations. In these populations, we used the same protocol as in the 'long-term control', but initial population size (5 males and 5 females) and number of generations (n = 31) was the same as in the 'stop-selection' experiment.

## Measuring viability selection on body size

We measured linear selection differentials (S) for female body weight (i.e. population mean dry weight after – population mean dry weight before the non-breeding season) in all three experiments ('long-term control', 'stop-selection' and 'short-term control'). We used female dry body weight as a proxy for body size. We experimentally confirmed that the mean body size declined along with mean dry weight in surviving females after the non-breeding season by placing individuals from the stock population in the non-breeding season in either high (n = 300) or low (n = 20) abundance. We measured the thorax size in 10 individual females sampled before the non-breeding season and 10 females sampled from the population after the non-breeding season. Mean thorax size after the non-breeding season was significantly lower than mean thorax size prior to the non-

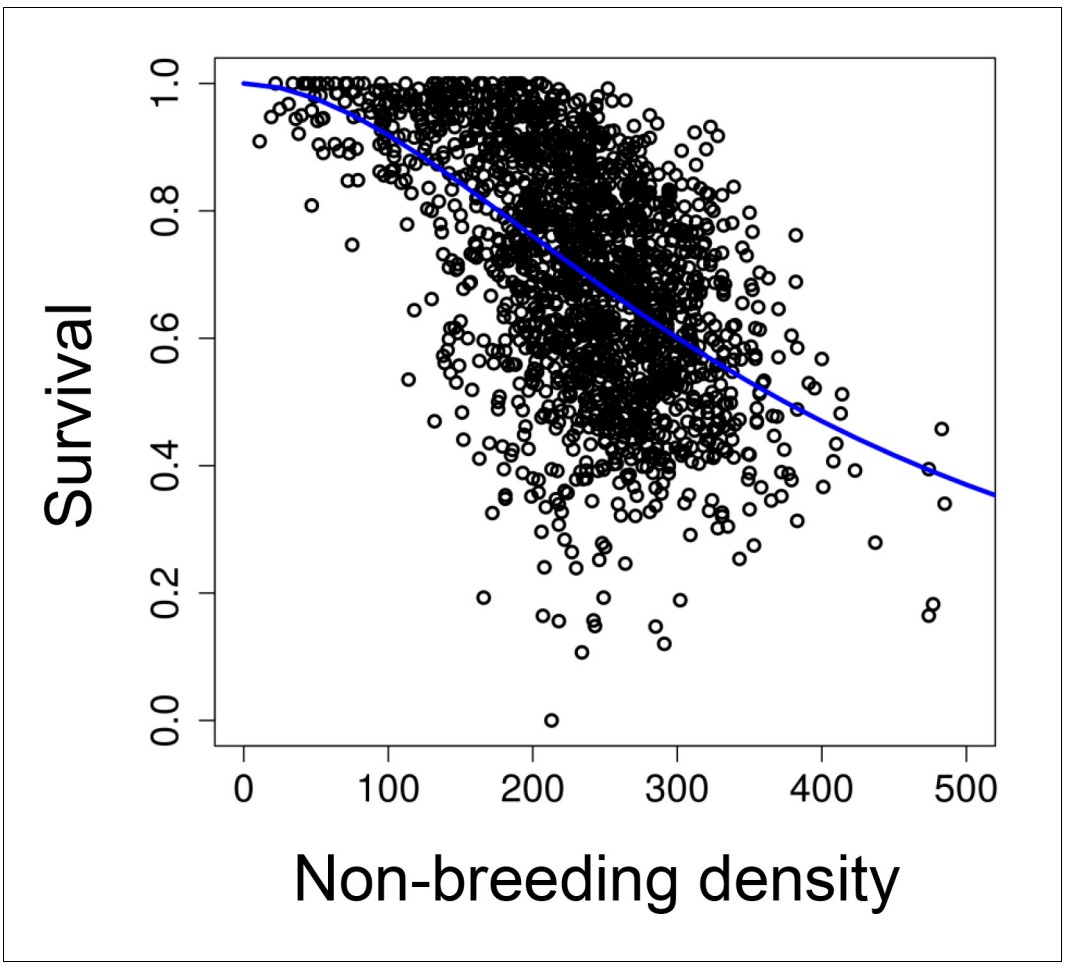

**Figure 6.** The relationship between survival and population size at the beginning of the non-breeding season. The blue line represents a survival function parameterized with our 'long-term control' populations ($v$ = 375.22, s.e. =7.60, $t$ = 49.43, p<0.001; $w$ = 1.83, s.e.=0.08, $t$ = 22.50, p<0.001).

breeding season (Welch t-test: $t$ = −2.87, df = 37.66, p=0.006), whereas there was no significant differences in average thorax in the low abundance treatment (Welch t-test: $t$ = −0.55, df = 36.96, p=0.586; average size before = 1.318 mm; average size after the non-breeding season for the high abundance treatment was 1.225 and 1.297 mm for the low abundance treatment).

We obtained female dry weight from 5% of total population size before and after the non-breeding season. We then sexed, dried and weighed individual females to the nearest 0.001 mg using a microbalance. To calculate the linear selection differentials (S) for female body weight, we standardized dry weight measures to a mean of zero and unit variance prior to calculating selection (*Brodie et al., 1995*). The significance of our estimates of viability selection during the non-breeding season was assessed using a Welch t-test that tested for differences between mean female dry body weight before and after the non-breeding season. We then used a linear mixed effect model (LMM) to investigate whether S was density-dependent. In the LMM, the selection differential calculated for each population in each generation was the response variable, number of individuals at the beginning of the non-breeding season was fitted as a fixed effect and population was fitted as a random effect.

In the 'long-term control', we measured female dry weight in 38 generations from an arbitrary number of replicate populations (16 to 25 population per generation) in generations 9, 10, 15, 16, 21, 22, 25, 26, 28–58. Total number of individuals measured was 5258 before and 3620 after the non-breeding season. In the 'stop-selection' and 'short-term control', we sampled females in seven

populations in generations 1 to 25 ($n$ = 1591 females and $n$ = 1979 females in the 'short-term control' and 'stop selection' treatments, respectively), with the exception of generation 13 in the 'short term control'.

## Documenting multigenerational cycles

To investigate whether population size cycled in our experimental populations, we used the autocorrelation function method (*Turchin and Taylor, 1992*; *Box et al., 2013*). Cycles can be inferred by investigating the shape of the estimated ACF: population cycles are characterized by stationary periodic dynamics, which result in an oscillatory decay to zero of the ACF estimates. We calculated the ACF for each treatment using the mean population size across all replicates at the start of the breeding and non-breeding season in each generation. For all-time series, we excluded the first three generations to avoid any transient effects caused by the initial population size. We also detrended the time series to eliminate any overall temporal trends in abundance by subtracting the fitted values from a linear regression of average population size on generation ('long-term control': $\beta$ = 0.468, s. e.=0.177, $t$ = −2.65, p=0.011; 'short-term control': $\beta$ = −1.855, s.e.=0.686, $t$ = −2.71, p=0.01).

To investigate temporal changes in body size in the 'long-term control' and 'stop-selection' populations (measured as female dry weight), we used the same autocorrelation function method described above for the population size, including detrending the data.

## Documenting delayed density dependence

We investigated the effects of past abundance on fecundity and survival with LMM with vial (population) as a random effect in all three experiments: 'long-term control' and 'short-term control' and 'stop-selection' treatment. In previous work, we have shown that fecundity in this system is influenced by an interaction between current abundance at the beginning of the breeding season and at the beginning of the previous non-breeding season (*Betini et al., 2013a*, *2013b*). Thus, in this study, we also investigated if this interaction significantly explained variation in fecundity in all three experiments. Fecundity was calculated as the number of individuals at the end of the breeding season (i.e. number of offspring) divided by the number of individuals at the beginning of the same breeding season (number of parents). Because we did not know how environmental effects (e.g. physiological condition) could influence survival in our system, we incorporated the breeding and non-breeding abundance going back two generations as explanatory variables in the model to explain survival. Lag effects have been documented in natural populations subjected to strong seasonality (*Merritt et al., 2001*, *2001*; *Stenseth et al., 2003*). Survival was calculated as the number of individuals at the end of the non-breeding season divided by the number of individuals at the beginning of the non-breeding season. Explanatory variables were standardized before analysis by subtracting the sample mean from each observation and dividing each value by the sample standard deviation and response variables were log transformed.

## Mathematical model

We developed a mathematical model to investigate the contributions of both viability selection and delayed density dependence to population dynamics. The model linked quantitative trait evolution with demography and was similar to integral projection modelling approaches that are commonly used to study ecological and evolutionary processes concurrently (*Slatkin, 1979*, *1980*; *Ellner and Rees, 2006*). In particular, the number of individuals with body size y at the next time step t + 1, n (y, t + 1), was a function of a projection function k (y) and n (y, t), such that n (y, t + 1) = k(y) * n (y, t). The projection function k (y) accounted for the additive genetic and environmental contributions to body size, as well as both density-independent and density-dependent differences in fecundity or survivourship among body sizes, depending on season.

## Model description

Two population sizes were modeled in discrete time: population size at the beginning of the breeding season, and population size at the beginning of the non-breeding season within each generation. For each season, the mean breeding value for body size and the mean phenotypic value for body size were modelled, where the phenotypic value of an individual was its breeding value plus a random environmental effect. We assumed that breeding and phenotypic values were normally

distributed and that the variance in breeding values and the variance in environmental effects remained constant across generations. The assumption of a constant variance in breeding values assumed that mutational variance and changes in genetic architecture (including linkage and epistatic interactions) could restore variance that was initially depleted by selection. Slatkin's model provides for changes in the variance of breeding values due to selection and considers the effect of genetic architecture, such as linkage (*Slatkin, 1979*, *1980*). In our experiment, we do not have information on the genetic architecture for body size, nor mutational variance. We did not see a decline in the magnitudes of the rates of change in body size in the experiment, which is consistent with the maintenance of heritability.

The fecundity of an individual during the breeding season was a function of its body size and current and past effects of density. Past density influenced fecundity via changes in the physiological conditions of individuals that spent the previous non-breeding season at high densities but survived to breed at low densities (i.e. a carry-over effect that was modelled as the interaction between density at the end of the breeding season in the previous generation and current breeding density (*Betini et al., 2013a*, *2014*). Current density could also negatively influence fecundity via density-dependent effects. There was a stronger negative effect of density on fecundity for large versus small individuals, although large individuals have higher intrinsic fecundity (*Figure 3* in the main text). During the 'stop selection' phase of the simulations, we decreased carry-over effects (parameters $\varphi_5$ and $\varphi_6$) by 25% given that flies had unlimited amount of food in the 'stop selection' treatments and were likely to be in better physiological condition than those in the 'short-term control' treatments. Parameters used in the fecundity function were chosen such that fecundities as a function of body size and densities were within the normal range observed in *D. melanogaster*.

The survival of an individual during the non-breeding season was also a function of its body size (when viability selection occurs) and current and past effects of density. As indicated by our previous experiment, larger flies had lower survival and this effect was magnified by increase density (*Betini et al., 2014*). We included the effects of past density beyond the ones described above, going back two generations. During the 'stop selection' phase of the simulations, survival was not a function of body size nor past density, but was instead only affected by current density, following the logistic equation Su for survivorship and the experimental design. To accomplish this and to ensure proper scaling of survival, we assigned all individuals the same phenotype in the survivorship function. Parameters for the survival function were based on values obtained in the long-term control (*Figure 2a* in the main text).

In the context of an individual's breeding and phenotypic values, an individual with breeding value x had an environmental effect y added with a mean effect that was a negative linear function of density and was normally distributed with variance V[e]. Consequently, each breeding value expressed a distribution of phenotypes as a function of the current environment.

In the context of survivorship, since each breeding value expressed a distribution of phenotypes, each breeding value expressed a distribution of survivorship and the model integrated across environmental effects to get the mean survivorship associated with a breeding value. This integration of survivorship across environmental effects gave the value of the projection function for a given breeding value. The projection function in the context of fecundity was modelled similarly; breeding values were transformed into phenotypes via a distribution of environmental effects, which when integrated gave the mean fecundity of individuals with a particular breeding value for body size.

Population size across seasons was the product of the current population size times the average fecundity (breeding season) or average survivorship (non breeding season) at the phenotypic level. Below is a list of variables, parameters and functions used in the model.

## Variables

$X_i$ - population size at the beginning of the breeding season *i* generations ago
$Y_i$ - population size at the beginning of the non-breeding season *i* generations ago
$\bar{b}_X$ - mean breeding value for body size at the beginning of the breeding season
$\bar{b}_Y$ - mean breeding value for body size at the beginning of the non-breeding season
$\bar{p}_X$ - mean phenotype for body size at the beginning of the breeding season
$\bar{p}_Y$ - mean phenotype for body size at the beginning of the non-breeding season
$z$ - body size

Note, a generation consists of a breeding season and then a non-breeding season. At the beginning of the breeding season the mean breeding value $(\bar{b}_X)$ is a function of densities in previous generations, such that i > 0.

## Parameters

$V_A = 0.003$ - additive genetic variance for body size
$V_E = 0.007$ - environmental variance for body size
  such that $h^2 = 0.30$ (**Prout and Barker, 1989**)
Fecundity function:

$$f\left(z, \overrightarrow{X}, \overrightarrow{Y}\right) = \left(\varphi_1 + (\varphi_2 z)^4\right) \exp\left(-(\varphi_3 + \varphi_4 z^4)(X_0 + \varphi_5 Y_1 + \varphi_6 X_0 Y_1)\right)$$

Survival function:

$$s\left(z, \overrightarrow{X}, \overrightarrow{Y}\right) = \frac{1 - z(\upsilon_1(X_1 + Y_1) + \upsilon_2(X_2 + Y_2))}{1 + \left(\frac{z\left(Y_0 + (\upsilon_3 Y_0 z)^{\exp(z)^{\upsilon_4}}\right)}{\upsilon_6}\right)^{\upsilon_5}}$$

Integral projection model:

$$\bar{b}_x = \frac{\int_{z_{min}}^{z_{max}} \int_{z_{min}-x}^{z_{max}-x} x s\left(x+y, \overrightarrow{X}, \overrightarrow{Y}\right) N(x, \bar{b}_Y, V_B) + N(y, e_X(X_1), V_E) dy dx}{\int_{z_{min}}^{z_{max}} \int_{z_{min}-x}^{z_{max}-x} s\left(x+y, \overrightarrow{X}, \overrightarrow{Y}\right) N(x, \bar{b}_Y, V_B) + N(y, e_X(X_1), V_E) dy dx}$$

$$X_0 = \frac{Y_1 \int_{z_{min}}^{z_{max}} s\left(y, \overrightarrow{X}, \overrightarrow{Y}\right) N(y, \bar{p}_Y, V_B + V_E) dy}{\int_{z_{min}}^{z_{max}} N(y, \bar{p}_Y, V_B + V_E) dy}$$

$$\bar{p}_x = \frac{\int_{z_{min}}^{z_{max}} \int_{z_{min}-x}^{z_{max}-x} (x+y) N(x, \bar{b}_Y, V_B) N(y, e_Y(Y_1), V_E) dy dx}{\int_{z_{min}}^{z_{max}} \int_{z_{min}-x}^{z_{max}-x} N(x, \bar{b}_X, V_B) N(y, e_Y(Y_1), V_E) dy dx}$$

$$\bar{b}_y = \frac{\int_{z_{min}}^{z_{max}} \int_{z_{min}-x}^{z_{max}-x} x f\left(x+y, \overrightarrow{X}, \overrightarrow{Y}\right) N(x, \bar{b}_X, V_B) N(y, e_Y(Y_1), V_E) dy dx}{\int_{z_{min}}^{z_{max}} \int_{z_{min}-x}^{z_{max}-x} f\left(x+y, \overrightarrow{X}, \overrightarrow{Y}\right) N(x, \bar{b}_X, V_B) N(y, e_Y(Y_1), V_E) dy dx}$$

$$Y_0 = \frac{X_0 \int_{z_{min}}^{z_{max}} f\left(y, \overrightarrow{X}, \overrightarrow{Y}\right) N(y, \bar{p}_X, V_B + V_E) dy}{\int_{z_{min}}^{z_{max}} N(y, \bar{p}_X, V_B + V_E) dy}$$

$$\bar{p}_y = \frac{\int_{z_{min}}^{z_{max}} \int_{z_{min}-x}^{z_{max}-x} (x+y) N(x, \bar{b}_Y, V_B) N(y, e_X(X_1), V_E) dy dx}{\int_{z_{min}}^{z_{max}} \int_{z_{min}-x}^{z_{max}-x} N(x, \bar{b}_Y, V_B) N(y, e_X(X_1), V_E) dy dx}$$

In the equations above $N(z, m, v)$ is the probability density of a normally distributed random variable with a mean of $m$ and variance $v$. Functions $e_X(N) = -\lambda_X N$ and $e_Y(N) = -\lambda_Y N$ give the average environmental effect on the phenotype.

## Other parameters and their values

$\varphi_1 = 1.5$ (intrinsic fecundity for small flies)
$\varphi_2 = 2.8$ (rate of increase in fecundity with body size)
$\varphi_3 = 0.00007$ (baseline rate of decline in fecundity with density for small flies)
$\varphi_4 = 0.04588$ (rate of increase in the magnitude of the decline in fecundity as body size increases)
$\varphi_5 = 1$ (constant characterizing the effects of population size at the end of the breeding season one generation ago; i.e. carry-over effects)

$\varphi_6 = 0.001$ (constant characterizing the interaction between current population size and population size at the end of the breeding season one generation ago)

$\upsilon_1 = 0.005$ (rate of decline in survivorship due to population sizes one generation ago)

$\upsilon_2 = 0.0026$ (rate of decline in survivorship due to population sizes two generations ago)

$\upsilon_3 = 0.7$ (constant governing negative effect of the interaction between the current population size and body size)

$\upsilon_4 = 0.7$ (constant governing the shape of the negative effect of the interaction between current population size and body size on survivorship)

$\upsilon_5 = 6$ (a second constant governing the shape of the negative effect of the interaction between current population size and body size on survivorship)

$\upsilon_6 = 350$ (a third constant governing the shape of the negative effect of the interaction between current population size and body size on survivorship)

$\lambda_X = 0.00001$ (rate of decline in body size [with density], i.e. the environmental effect of density on body size due to density at the beginning of the breeding season)

$\lambda_Y = 0.00025$ (rate of decline in body size [with density], i.e. the environmental effect of density on body size due to density at the beginning of the non-breeding season)

All LMM were performed using the lmer function from the lme4 package (*Bates, 2010*) and p-values were obtained using the *lmerTest* package (*Kuznetsova et al., 2014*). All analyses were conducted in R (*R Core Team, 2015*). Marginal ($R^2_{LMM(m)}$) and conditional ($R^2_{L<MM(c)}$) variance for the LMMs were calculated with *MuMIn* package according to (*Nakagawa and Schielzeth, 2013*). $R^2_{LMM(m)}$ is the variance on the response variable that is explained only by the fixed effects and $R^2_{LMM(c)}$ is the variance that is explained by both fixed and random effects (*Nakagawa and Schielzeth, 2013*).

## Acknowledgements

We thank E Hentsch, RJ Kilgour, RS Cruz and AN Pessoa for assistance in the lab and three anonymous reviewers for helpful suggestions. The research was supported by NSERC Discovery grants to DRN, CKG, AGM, a University Research Chair from the University of Guelph and an Early Researcher Award to DRN and an Ontario Graduate Scholarship to GSB.

## Additional information

### Funding

| Funder | Author |
| --- | --- |
| Ontario Graduate Scholarship | Gustavo S Betini |
| Natural Sciences and Engineering Research Council of Canada | Andrew G McAdam<br>Cortland K Griswold<br>D Ryan Norris |

The funders had no role in study design, data collection and interpretation, or the decision to submit the work for publication.

### Author contributions

GSB, Conceptualization, Formal analysis, Investigation, Methodology, Writing—original draft, Writing—review and editing; AGM, Conceptualization, Supervision, Funding acquisition, Writing—original draft, Writing—review and editing; CKG, Conceptualization, Formal analysis, Supervision, Funding acquisition, Writing—original draft, Writing—review and editing; DRN, Conceptualization, Resources, Supervision, Funding acquisition, Writing—original draft, Writing—review and editing

### Author ORCIDs

Gustavo S Betini, http://orcid.org/0000-0003-0707-4128
Andrew G McAdam, http://orcid.org/0000-0001-7323-2572
D Ryan Norris, http://orcid.org/0000-0003-4874-1425

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

## Appendix 1

### Experimental system

We used *Drosophila melanogaster* from an outbred population collected in Dahomey (now Benin) in 1970, which has since been maintained in cage culture at 25°C. To simulate seasonality in populations with non-overlapping generations, we changed food composition to create two distinct 'seasons'. During the 'breeding season', flies were allowed to lay eggs for 24 hr (day 0) in a dead yeast–sugar medium (1000 ml $H_2O$, 100 g sucrose, 50 g Fleischmann's yeast, 16 g agar, 8 g $C_4H_4KNaO_6$, 1 g $KH_2PO_4$, 0.5 g NaCl, 0.5 g $MgCl_2$, 0.5 g $CaCl_2$, 0.5 g $Fe_2(SO_4)_3$), after which adults were discarded and larvae were allowed to mature to adults. During the breeding season, emerging offspring were moved from old to new vials containing fresh food on days 10, 12 and 14 to avoid high offspring mortality (**Dey and Joshi, 2006**). During this period, densities of offspring were not manipulated, that is, their densities were a function of the number of flies that emerged during the breeding season. Flies were marked with a fluorescent dust (as part of the protocol of another experiment) and let rest overnight in larger bottles (177.4 ml) with 20 ml of food so that they could easily remove the excess of dust. On day 17, flies were lightly anesthetized with $CO_2$, counted and placed into the non-breeding vials. The non-breeding season consisted of an empty vial of the same size as the breeding vials ($28 \times 95$ mm) and a pipette tip filled with 0.200 ml of 5% water–sugar solution per day. This medium prevented females from producing eggs (**Terashima et al., 2005**) but also resulted in density-dependent mortality, as would be expected from wild populations (**Betini et al., 2013a**). Oviposition resumed within <12 hr when flies were placed back on a protein-rich food (**Terashima et al., 2005**). After 4 days in the non-breeding season, surviving flies were counted and the cycle was repeated. Note that since adults were removed after laying at the start of the breeding season, there were non-overlapping generations and each generation experienced only a single breeding and non-breeding season. We also randomly removed 5% of each population each time they were moved between seasons to mimic migratory mortality and dispersal. This procedure was repeated for 58 generations in each of 45 replicate populations. During all experiments, flies were kept at 25°C, 12 hr light/dark cycles and humidity was between 30% to 50%.

As in many other animals, body size was positively related to fecundity in our experimental system according to a linear mixed effect model with per capita breeding output as the response variable, female dry weight as a explanatory variable (average for each replicate, over all 38 generations) and population as a random effects (**Appendix 1—figure 1** and **Appendix 1—table 1**).

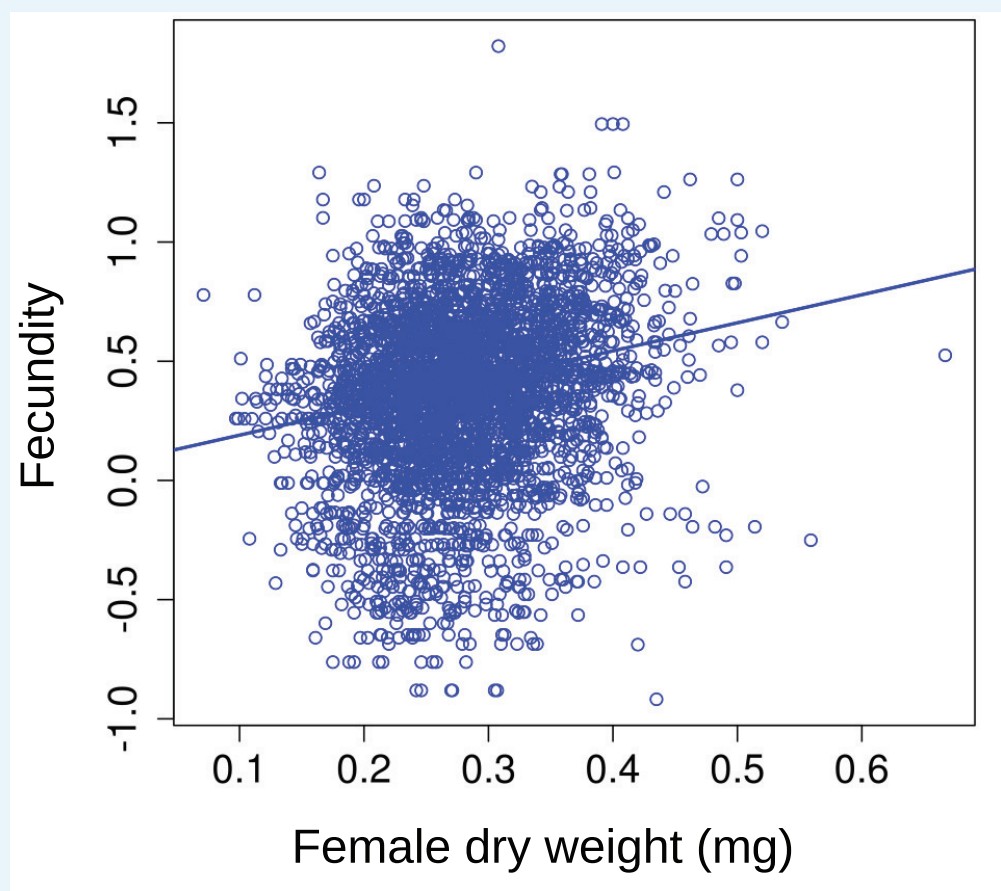

**Appendix 1—figure 1.** Fecundity as a function of average female dry weight at the beginning of the breeding season. Larger individuals tend to have higher fecundity. The solid line represents the linear mixed effect model line for female dry weight (*Appendix 1—table 1*).

**Appendix 1—table 1.** Parameter estimates obtained from a linear mixed effect model with fecundity as a response variable and average female dry weight at the beginning of the breeding season as explanatory variable. $R^2_{LMM(m)}$=0.04 and $R^2_{LMM(c)}$=0.14.

| Parameters | | SE | df | t | p |
|---|---|---|---|---|---|
| | Fixed effects estimate | | | | |
| Intercept | 0.072 | 0.043 | 260 | 1.69 | 0.101 |
| Female dry weight | 1.178 | 0.092 | 3601 | 12.82 | <0.001 |
| | Random effect variance | | | | |
| Population | 0.015 | | | | |
| Residual | 0.125 | | | | |

**Appendix 2**

## Multigenerational cycles

To understand the overall effect of seasonality on changes in population size over time, we compared the time series generated by our 'long-term control' with another treatment where 30 replicate populations of flies were maintained in breeding vials (28 × 95 mm) with standard medium across 30 generations. These populations lacked a non-breeding period, so we refer to these as the 'aseasonal' populations (*Appendix 2—figure 1a,b*). In these populations, all offspring were moved to a new fresh vials with breeding medium on day 16.

To investigate whether population size cycled in the 'aseasonal' treatment, we used the same autocorrelation function method described in the main text, after detrending the data ('aseasonal': $\beta$ = 0.403, s.e.=0.579, $t$ = 0.69, p=0.49; *Appendix 2—figure 1*).

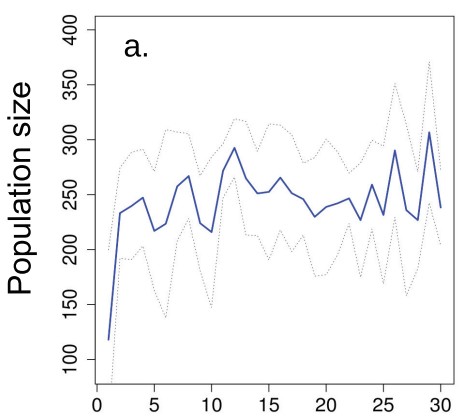
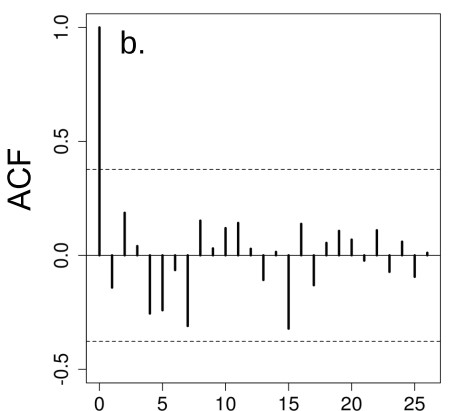

**Appendix 2—figure 1.** Overall effect of seasonality in multigenerational cycles. (**a**) Population size of 'aseasonal' populations over 30 generations do not present evidence of multigenerational cycles according to the (**b**) autocorrelation plots. In *a* solid blue line denotes mean population size for each generation from all replicates and dotted lines denote ±1 s.d.

## Viability selection

We found that viability selection on body size ($S$) became more negative (i.e. stronger selection for reduce size) with increasing non-breeding density (*Figure 2c* in the main text; *Appendix 3—table 1*) and this was driven by changes in mean dry weight after the non-breeding season rather than changes in the mean dry weight before the non-breeding season (*Appendix 3—table 1*; *Appendix 3—figure 1*).

**Appendix 3—table 1.** Parameter estimates obtained from a linear mixed effect models to investigate variation in female dry weight as a function of the interaction between density at the beginning of the non-breeding season and whether weight was measured before or after the non-breeding season. Status refers to a dummy variable indicating the time when body size was measured (before or after the non-breeding season). Reference value was after the non-breeding season. Marginal (only taking account the fixed effects) and conditional (taking into account both fixed and random effects) coefficient of determination were $R^2_{LMM(m)}$=0.42 and $R^2_{LMM(c)}$=0.50, respectively. In all models, population and generation were entered as random effects.

| Parameters | | SE | dfDf | tT | pP |
|---|---|---|---|---|---|
| | Fixed effects estimate | | | | |
| Intercept | 0.680 | 0.06 | 266 | 10.31 | <0.001 |
| Non-breeding density | 0.005 | 0.0002 | 900 | −29.38 | <0.001 |
| Status Before | 0.430 | 0.009 | 896 | 7.55 | <0.001 |
| Non-breeding density:Status | 0.003 | 0.0001 | 892 | 14.11 | <0.001 |
| | Random effect variance | SD | | | |
| Generation | 0.045 | 0.212 | | | |
| Population | 0.037 | 0.193 | | | |
| Residual | 0.474 | 0.688 | | | |

Betini *et al.* eLife 2017;6:e18770. DOI: 10.7554/eLife.18770 

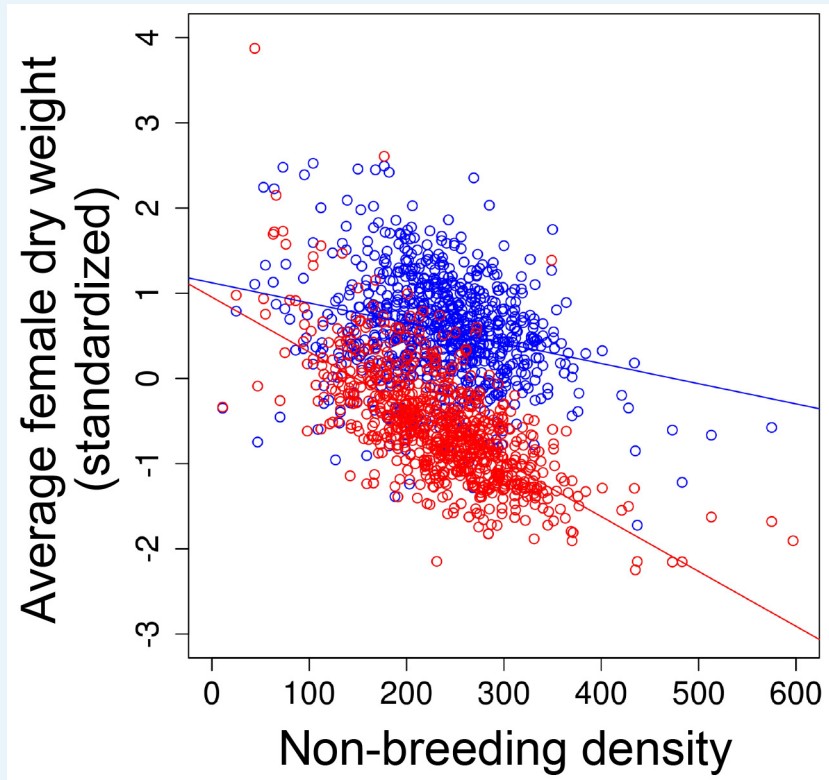

**Appendix 3—figure 1.** The strength of selection on body weight in seasonal environments. As non-breeding density increased, the magnitude of directional selection for reduced dry weight also increased (*Figure 2c* in the main text). This relationship was driven by changes in the weight after (red points) rather than before (blue points) the non-breeding season. Solid lines indicate the linear mixed effect model line for non-breeding density before (blue line) and after (red line) the non-breeding season.

