## [Decision Letter]

[Editors’ note: a previous version of this study was rejected after peer review, but the authors submitted for reconsideration. The first decision letter after peer review is shown below.]

Thank you for submitting your work entitled "Fitness trade-off between seasons causes multigenerational cycles in phenotype and population size" for consideration by *eLife*. Your article has been reviewed by two peer reviewers, one of whom, Lutz Becks (Reviewer #1), is a member of our Board of Reviewing Editors, and the evaluation has been overseen by a Senior Editor.

Our decision has been reached after consultation between the reviewers. Based on these discussions and the individual reviews below, we regret to inform you that your work will not be considered further for publication in *eLife*.

While the reviewers agreed that this is an interesting manuscript with a timely topic and interesting results, the reviewers also felt that the experiments reported in the manuscript don't provide sufficiently strong support for the conclusions. Some of the interpretation of the results are not supported by statistical tests and the study lacks an effort to establish lines at different time points to determine whether genetic changes did occur. The overall structure of the manuscript was confusing. In summary, we didn't feel that the manuscript provides a decisive answer to the question raised.

Reviewer #1:

This is an interesting study examining the joined effects of ecological and evolutionary processes on population dynamics in seasonally fluctuating environments. Using a series of experimental evolution studies with *Drosophila* populations, Betini and colleagues show how density dependency and viability selection leads to multi-generational cycles of the populations. They make use of the fact that body size affects fecundity (positively) and survival (negatively) and that these two traits trade off between breeding and non-breeding seasons (limited food supply). Specifically, they compare the evolutionary (body size) and ecological changes (density) within populations when manipulating seasonality (absence, presence with very low and low food supply). They found that seasonality leads to multi-generational cycles of 7-9 generations and that these cycles increase with lower food supply in the non-breeding season. An additional mathematical model shows that the multi-generational cycles depend on the trade-off and differential selection between seasons and the density-dependency of growth. Overall, this is an important study showing in an elegant way the underlying mechanism producing multi-generational cycles in populations without interspecific species interactions. It adds to the growing evidence that ecological and (rapid) evolutionary dynamics interact at one timescale. I have however several concerns that need to be addressed:

1) The Results section is not clearly structured and the differences between the different treatments were only clear after reading it several times. I suggest to restructure this section and provide the reader with a short overview of what has been tested in which experiment at the beginning of this section, which should also mention the predictions.

2) Some statements are not supported by statistical analyses and it is difficult to judge the results:

– The fold changes in population size differences within the multi-generational cycles are not supported by any statistical analyses and for the 'long-term-control' not easy to see in the data. Could you provide some statistical test for the differences in amplitude?

– Figure 7: the ACF plot does not provide evidence for a statistical period in the time series.

– Could you provide some statistical evidence for the model results (e.g., AFC plots similar to the experimental ones) so the reader can compare better?)?

- Subsection “Mathematical Model”: how do you judge 'unstable population dynamics' in your model?

– It is interesting to see that the period of the densities (Figure 2) and female dry weight (Figure 6) are very similar (as expected). Is there a correlation between these two time series? And what would one expect?

3) The model should be introduced in the main text for the reader to be able to follow better what was manipulated and to test for the effects of viability selection and delayed density effects.

4) ACF analyses: please provide more explanation for the ACF plots, so that the readers know how to interpret the maximum in ACF and the significance line you show in the plots.

Reviewer #2:

I am deeply interested in the question and I found the work to be done well. However, I cannot fully judge the novelty of the work within the ecological literature. The eco-evolutionary part of the work is less well developed as it is not fully clear whether the evolutionary changes were really important in the generation of the longer cycles as the authors claim even though they have some suggestive data. At the end, lack of effort to establish lines at different times of the cycles and determine whether genetic changes did occur I think limits the work. My hunch is that it does not rise to the level of an *eLife* paper.

[Editors’ note: what now follows is the decision letter after the authors submitted for further consideration.]

Thank you for choosing to send your work entitled "Fitness trade-off between seasons causes multigenerational cycles in phenotype and population size" for consideration at *eLife*. Your letter of appeal has been considered by a Senior Editor, a Reviewing editor, and one reviewer and we are prepared to consider a revised submission with no guarantees of acceptance.

The Reviewing Editor has drafted this decision to help you prepare a revised submission.

Reviewer #3:

The authors present an impressive set of long lasting and well replicated experiments testing important eco-evolutionary consequences of seasonality. This shows the power of model lab systems! I think the questions addressed were well introduced building up from more targeted mechanisms to more complex eco-evo interactions that could possibly explain well known ecological patterns (multigenerational populations cycles).

My main concern is that the manuscript is overly complicated mostly due to a lack of proper organization. It was very confusing reading this for the first and even the second time. I have made a number of suggestions below that reorder the methods and improve clarity. The interesting manipulation to compare the control to the stop-selection experiment is lost in the confusion that is the methods sections.

Main Concerns:

1) There is a lot of importance given to delayed density dependence in the Discussion (but none in the Introduction). However, there was no formal test of it, which I could recognize, in the actual empirical data even though the model suggests it is very important. I am not an expert in this topic so why wasn't the experimental data tested for it?

2) The manuscript was very difficult to follow and unnecessarily confusing. I think this stems from two many issues.

a) The number, naming, and ordering of experiments is confusing. There are a lot of similar experiments being presented and it is difficult to follow which is being discussed or how they differ. This is caused firstly by the having their descriptions only appear after the results (see point 2). In addition the order of the experiments doesn't seem to match the figures or results. Given the number of experiments it might be helpful to replace one of the many results graphs with a simple illustration showing the various experiments. b) Because the journal format is to put methods after the results (I am not a fan of this) the authors need to adjust their sections to reflect this or else the manuscript is unintelligible if read the way the journal intends. This means that results and figure captions need to be understandable before getting to the methods. Currently, both of these sections refer to experiments that have not been described or even mentioned yet.

I think the most important thing to add to alleviate both issues a and b is to add a paragraph in the Introduction that says: To address question 1 we compared the following experiments: long-term seasonal control (brief description) versus stop-selection (brief description). We then test the prediction that if resources are reduced, the trade-off should magnify the effect leading to.…. We tested this by.…. and so on. In this summary use the same terminology as the rest of the paper. Currently the Introduction fails to introduce half of the experiments.

The Introduction should also mention the importance of delayed density dependence in a bit more detail than in half a sentence.

Methods: The methods seem to be in a random order.

I would restructure the methods to help readability. Start with 5 small paragraphs one for each experiment subtitled by the experiment name. Please include the short-term control in the methods also. The order of the experiments should match the figures, results, and questions. Then discuss data collection and the analyses (ACF… stats…).

Subsection “Multigenerational cycles”, paragraph two, his comes out of nowhere. I though the prediction is being tested by a treatment with a non-breeding season but no selection.

Where all the 5 experiments conducted at the same time? If not there could be differences in the lab environment among experiments that could possibly (although probably unlikely) impact the multigenerational dynamics.

Results: The results jump among experiments and it is unclear which experiments are being discussed. Maybe it is best to explain what happened in the Control Experiment (all results). Then compare it to the other experiments.

At all the times the manuscript should refer to the experiments by their established names, e.g subsection “Long-term control” should say the aseasonal experiment. Why is this result presented before the no-selection experiment? Before Figure 2?

In the same paragraph there are experiments that were never introduced

Paragraph two? For which experiment? the long or short term?

Subsection “Short-term control”. I don't think initial population size of the experiments is mentioned anywhere except here in the results. Why present a complication concerning the stop-selection experiment before presenting the stop-selection experiment or their results?

Subsection “’Stop selection’ experiment following the order of these figures is exhausting and confusing.

Figures:

Titles don't match content (see Figure 3 and Figure 6 for example).

Consider a large figure that includes all experiments (combines Figure 2 and Figure 3 and Figure 7). If the 4 other experiments are all compared to the long-term control then they should all be in the same figure.

Discussion

The Discussion is only big picture and skips any discussion of many of the results (no mention of the reduced food experiment results at all….) This disconnects the work from the bigger interpretation and feels unbalanced.

Discussion paragraph two, I am not sure what 'at ecological scales' means Do you mean timescales? If so there are numerous reviews with hundreds of examples of rapid evolution of ecologically relevant traits (Hendry, Kinison, Reznick, Thompson…). Whether these have been robustly shown to impact ecological dynamics is less clear but see good experimental examples by Willliams Levine 2016, Turcotte et al. 2011 or 2103, Becks multiple, Yoshida et al. 2003 and numerous others by Hairston).

A lot of the Discussion concerns delayed density dependence but this was barely mentioned in the Introduction.

Immediately when I saw the title of the manuscript, and especially the Abstract, I thought of Paul Schmidt's work (U Penn) on seasonal adaptation in *Drosophila*. Although the authors cited one of their studies I think the links to this body of work should be more prominent as they have discussed evolutionary changes over multiple generations in a lab experiment that are driven by life-history trade-offs (Schmidt Conde 2006, Evolution).

[Editors' note: further revisions were requested prior to acceptance, as described below.]

Thank you for resubmitting your work entitled "A fitness trade-off between seasons causes multigenerational cycles in phenotype and population size" for further consideration at *eLife*. Your revised article has been favorably evaluated by Ian Baldwin (Senior editor) and a Reviewing editor.

The manuscript has been improved but there are some remaining issues that need to be addressed before acceptance, as outlined below:

1) The revised manuscript includes comments and changes in response to reviewer 3, but the comments by reviewer 1 and 2 have not been taken into consideration when revising the manuscript (although commented on when appealing the first decision). The authors should make those changes as well.

2) In addition, the revised version needs further restructuring and clarification as it is still not clear from reading what experiments have been done and why (e.g., add an overview figure).

3) Also, in the paragraph 'Delayed density dependence' of the Results section, the authors compare the results of independent correlations between selection differentials and population sizes. A formal statistical test such as ANCOVA or mixed effect model is need for this comparison and the authors have all the data.

---

## [Author Response]

[Editors’ note: the author responses to the first round of peer review follow.]

While the reviewers agreed that this is an interesting manuscript with a timely topic and interesting results, the reviewers also felt that the experiments reported in the manuscript don't provide sufficiently strong support for the conclusions.

We understand the point made by the Editor here but we don’t see any specific comments by the reviewers that indicate that our experiments do not support our conclusions. It seems like reviewer 1 has questioned some of the statistical approaches and addressing these comments would require minor clarifications.

Some of the interpretation of the results are not supported by statistical tests and the study lacks an effort to establish lines at different time points to determine whether genetic changes did occur.

The first point here, that the results are not supported by the statistical tests, arises from several minor comments provided by reviewer 1. Our interpretations of the results are well supported by all of our statistical tests. The second point, made by reviewer 2, has been addressed in detail below but to summarize here: our study shows that cycles can be generated by fluctuating episodes of natural selection and that genetic change (i.e. an evolutionary response to this selection) is not required for population cycles to emerge. This is, in fact, one of the novel aspects of our work.

The overall structure of the manuscript was confusing.

We have made several changes in the manuscript to address this issue (see specific points in the response to the reviewers).

In summary, we didn't feel that the manuscript provides a decisive answer to the question raised.

We respectfully disagree with this point. We asked whether a seasonal fitness trade-off could affect changes in population size. We showed empirically that both aseasonal populations and populations with no or weak viability selection in the non-breeding season had very different population dynamics and phenotypic variation compared to controls. In addition, our mathematical model supports our empirical results. As reported in the paper, when heritability is

0.3 for body size (equal to published estimates) and natural selection occurs in the non-breeding season (differential survival), the model generates cycles in population size. But if we ‘turn-off’ natural selection in the non-breeding season, then the population crashes. Furthermore, the model also simulates a small average decline in body size with time, as seen in the long-term control treatments. Importantly, it is also the case that if heritability is nearly zero (*h^2^*=10^-5^) and natural selection occurs in the non-breeding season, the model generates cycles in population size. This result clearly shows that an evolutionary response to selection is not required for population cycles to emerge. Likewise, when we turn off natural selection in the non-breeding season, there are no cycles and the population crashes. When the heritability of body size is nearly zero there is no average decline in body size, which is expected when there is no between generation response to selection.

For simplicity, we focused on results when *h^2^=0.3* because this reproduced all aspects of the trajectories of population size with time. We have now added the predicted population size based on the mathematical model when heritability is close to zero (Figure 5).

Reviewer #1:

This is an interesting study examining the joined effects of ecological and evolutionary processes on population dynamics in seasonally fluctuating environments. Using a series of experimental evolution studies with Drosophila populations, Betini and colleagues show how density dependency and viability selection leads to multi-generational cycles of the populations. They make use of the fact that body size affects fecundity (positively) and survival (negatively) and that these two traits trade off between breeding and non-breeding seasons (limited food supply). Specifically, they compare the evolutionary (body size) and ecological changes (density) within populations when manipulating seasonality (absence, presence with very low and low food supply). They found that seasonality leads to multi-generational cycles of 7-9 generations and that these cycles increase with lower food supply in the non-breeding season. An additional mathematical model shows that the multi-generational cycles depend on the trade-off and differential selection between seasons and the density-dependency of growth. Overall, this is an important study showing in an elegant way the underlying mechanism producing multi-generational cycles in populations without interspecific species interactions. It adds to the growing evidence that ecological and (rapid) evolutionary dynamics interact at one timescale. I have however several concerns that need to be addressed:

1) The Results section is not clearly structured and the differences between the different treatments were only clear after reading it several times. I suggest to restructure this section and provide the reader with a short overview of what has been tested in which experiment at the beginning of this section, which should also mention the predictions.

We have made several changes in the manuscript to improve clarity. For example, we have now added a new paragraph in the Introduction to describe the experiments and link them with our predictions. Given that the methods section comes after the presentation of the results, we have also clearly explained the goal of each analysis when presenting the results. For example, on we wrote: “To investigate the presence of viability selection for small body size and whether this selection was density-dependent, we measured female dry weight in 38 generations from 25 different populations.”. Further on we wrote: “We statistically investigated whether fecundity and survival were influenced by densities in past seasons (i.e. delayed density dependence) in all three experiments: ‘long-term control’ and ‘short-term control’ and ‘stop-selection’”.

Furthermore, following the suggestion of the editor, we have also included a new figure that consists of a schematic of the experiments (now Figure 1), as well as the length and the purpose of each experiment.

2) Some statements are not supported by statistical analyses and it is difficult to judge the results:

– The fold changes in population size differences within the multi-generational cycles are not supported by any statistical analyses and for the 'long-term-control' not easy to see in the data. Could you provide some statistical test for the differences in amplitude?

As suggested by reviewer 1, we have modified the way we present the time series for all three experiments (please see Figure 2, Figure 3 and Figure 4 in the main text) to help the reader evaluate these differences in the data. However, we are not aware of a formal statistical test to compare amplitudes. We could use an arbitrary cut-off value and apply a simple t-test. For example, we could take the 5% highest (>281, mean = 299) and lowest (<132, mean = 122) values in the time series and run a simple Welch Two Sample t-test. In the case of the long-term control treatment, there is a statistical difference between the two means (t = -21.152, df = 9.87, p < 0.001). This result is robust for a wide range of cut-off values. However, as we indicated, these are arbitrary cut-offs and we would rather avoid using them.

– Figure 7: the ACF plot does not provide evidence for a statistical period in the time series.

We understand the concerns raised by reviewer 1 here, but we believe the important point about the ACF plots is the overall shape of the ACF estimates. In the long-term control, the ACF plot is characterized by an oscillatory decay to zero of the ACF estimates, which is an evidence of a stationary and periodic process (inset plot in Figure 2 in the main text; see our methods section and Turchin and Taylor 1992). This characterization of cyclic dynamic is lacking in the ACF estimates for the stop-selection treatment (inset plot in Figure 1 in the main text). In addition, there is no consensus in the literature about the relevance of statistical tests for periodicity. Different methods or different choices within methods could give different results. For example, in our manuscript, we have eliminated the first 3 generations to avoid transient dynamics. However, we could have eliminated the first 5 or 6 generations, which would result in statistical evidence for a 16 generation cycle in our long-term treatment, but still no evidence of cycle in the stop-selection treatment (see Figure 10 and Figure 11).

Author response image 1.ACF plot for the long-term control after eliminating the first 6 generations to avoid transient dynamics.Note that the peak at lag 16 crosses the confidence interval line.**DOI:**
http://dx.doi.org/10.7554/eLife.18770.017

Author response image 2.ACF plot for the stop-selection treatment after eliminating the first 6 generations to avoid transient dynamics.There is no evidence of periodicity.**DOI:**
http://dx.doi.org/10.7554/eLife.18770.018

– Could you provide some statistical evidence for the model results (e.g., AFC plots similar to the experimental ones) so the reader can compare better?)?

We have now added the time series for the model results (Figure 5).

– Subsection “Mathematical Model”: how do you judge 'unstable population dynamics' in your model?

When the population crashes to zero during simulations. We have now added this explanation to the main text: “This led to unstable population dynamics (i.e. the population crashed after 10 generations; Figure 5)”.

– It is interesting to see that the period of the densities (Figure 2) and female dry weight (Figure 6) are very similar (as expected). Is there a correlation between these two time series? And what would one expect?

The correlation between population size at the beginning of the non-breeding season at time t and female dry weight after the non-breeding season at time t is -0.64 (Pearson's product-moment correlation = -0.64; t = -4.48, df = 29, p-value < 0.001). This result supports the prediction from our hypothesis that high population size in the non-breeding causes an overall decline in body size. We have added this result to the main text (subsection “Long-term Control”).

3) The model should be introduced in the main text for the reader to be able to follow better what was manipulated and to test for the effects of viability selection and delayed density effects.

We have now moved the model description to the main text (subsection “Mathematical model”)

4) ACF analyses: please provide more explanation for the ACF plots, so that the readers know how to interpret the maximum in ACF and the significance line you show in the plots.

We have now provided more explanation for the ACF plots in both the Results and the analysis sections (”Documenting multigenerational cycles”). In the analysis section, we wrote: “Cycles can be inferred by investigating the shape of the estimated ACF: population cycles are characterized by stationary periodic dynamics, which result in an oscillatory decay to zero of the ACF estimates.”

Reviewer #2:

I am deeply interested in the question and I found the work to be done well. However, I cannot fully judge the novelty of the work within the ecological literature. The eco-evolutionary part of the work is less well developed as it is not fully clear whether the evolutionary changes were really important in the generation of the longer cycles as the authors claim even though they have some suggestive data. At the end, lack of effort to establish lines at different times of the cycles and determine whether genetic changes did occur I think limits the work. My hunch is that it does not rise to the level of an eLife paper.

From what we can infer from the overall nature of the comments, this is likely the primary reason why our paper was rejected. The reviewer’s feeling is that our paper does not rise to the level of an *eLife* paper because we have not shown genetic changes, which does not support the eco-evolutionary aspect of the study. We respectfully disagree with this viewpoint. We have shown that cycles can occur by internal processes through a combination of opposing episodes of natural selection between seasons (an evolutionary process) and delayed density dependence (an ecological process). Furthermore, our model shows that accompanying genetic changes are not necessary for these cycles to occur. From an ecological perspective, population cycles are typically thought to be driven by external processes, such as climate or predators, which, from an ecological perspective make our results very significant. Not only that, but we show that these cycles are driven by 3 very common elements found in almost all natural systems: seasonality, density-dependence, and natural selection, which makes our laboratory-based results applicable to wild species. Understandably, the reviewer admits that she/he is not well versed on the ecological significance of these results. However, we do not think that the paper should be rejected based on the grounds that we have not shown concurrent genetic changes because we have provided evidence that genetic changes are not necessary for cycles to occur. This represents a fundamental misunderstanding of our main finding, for which we take responsibility.

To address some of these issues, we have now provided the results with the model with low heritability in the main text, suggesting that cycles can occur in the presence of strong selection and in the absence of genetic changes across generations (subsection “Mathematical model”). We have also emphasized this in the Discussion.

[Editors' note: the author responses to the re-review follow.]

Reviewer #3:

The authors present an impressive set of long lasting and well replicated experiments testing important eco-evolutionary consequences of seasonality. This shows the power of model lab systems! I think the questions addressed were well introduced building up from more targeted mechanisms to more complex eco-evo interactions that could possibly explain well known ecological patterns (multigenerational populations cycles).

My main concern is that the manuscript is overly complicated mostly due to a lack of proper organization. It was very confusing reading this for the first and even the second time. I have made a number of suggestions below that reorder the methods and improve clarity. The interesting manipulation to compare the control to the stop-selection experiment is lost in the confusion that is the methods sections.

As suggested, we have now re-arranged the Introduction, Results and Materials and methods (see details below) and believe the clarity of the manuscript has improved substantially.

Main Concerns:

1) There is a lot of importance given to delayed density dependence in the Discussion (but none in the Introduction). However, there was no formal test of it, which I could recognize, in the actual empirical data even though the model suggests it is very important. I am not an expert in this topic so why wasn't the experimental data tested for it?

As suggested, we have now used mixed effect models to investigate the effect of past densities on fecundity and survival (“Short-term control” and “Documenting multigenerational cycles), and have reported the results in Table 2 and 3. In the Introduction, we have also clearly explained how delayed density dependence can influence population fluctuations (Last paragraph).

2) The manuscript was very difficult to follow and unnecessarily confusing. I think this stems from two many issues.

*a) The number, naming, and ordering of experiments is confusing. There are a lot of similar experiments being presented and it is difficult to follow which is being discussed or how they differ. This is caused firstly by the having their descriptions only appear after the results (see point 2). In addition the order of the experiments doesn't seem to match the figures or results. Given the number of experiments it might be helpful to replace one of the many results graphs with a simple illustration showing the various experiments. b) Because the journal format is to put methods after the results (I am not a fan of this) the authors need to adjust their sections to reflect this or else the manuscript is unintelligible if read the way the journal intends. This means that results and figure captions need to be understandable before getting to the methods. Currently, both of these sections refer to experiments that have not been described or even mentioned yet.*

I think the most important thing to add to alleviate both issues a and b is to add a paragraph in the Introduction that says: To address question 1 we compared the following experiments: long-term seasonal control (brief description) versus stop-selection (brief description). We then test the prediction that if resources are reduced, the trade-off should magnify the effect leading to.…. We tested this by.…. and so on. In this summary use the same terminology as the rest of the paper. Currently the Introduction fails to introduce half of the experiments.

We thank reviewer 3 for the helpful comments. We have now added a new paragraph in the Introduction to describe the experiments and link them with our predictions. Given that the methods section comes after the presentation of the results, we have also clearly explained the goal of each analysis when presenting the results.

The Introduction should also mention the importance of delayed density dependence in a bit more detail than in half a sentence.

We agree with reviewer 3 and have now explained how delayed density dependence could influence population fluctuations.

Methods: The methods seem to be in a random order.

I would restructure the methods to help readability. Start with 5 small paragraphs one for each experiment subtitled by the experiment name. Please include the short-term control in the methods also.

As suggested, we have modified the Materials and methods and included the description for the short-term control (subsection “’Stop-selection’ experiment”). We now introduced each experiment in the same order as presented in the Introduction and Results: long-term control, stop-selection and short-term control.

The order of the experiments should match the figures, results, and questions. Then discuss data collection and the analyses (ACF… stats…).

This is now fixed in the manuscript. In the Introduction, Results and Materials and methods, we first present the long-term control, then the stop-selection and finally short-term control.

Subsection “Multigenerational cycles”, paragraph two, this comes out of nowhere. I though the prediction is being tested by a treatment with a non-breeding season but no selection.

We are sorry for the confusion. These experiments were used to show that populations of *Drosophila* do not show multigenerational cycle under standard rearing conditions (aseasonal experiment). However, as reviewer 3 pointed out, the crucial comparison is between the long-term and the stop-selection. Thus, to improve clarity, we have moved the aseasonal experiment to the appendix. In addition, after re-organizing the results and methods based on the reviewer’s comments, we realized that the reduced-food experiment did not add any new information to our study, so we have decided to remove it from the manuscript.

Where all the 5 experiments conducted at the same time? If not there could be differences in the lab environment among experiments that could possibly (although probably unlikely) impact the multigenerational dynamics.

The stop-selection and short-term control experiments were conducted at the same time. The critical comparison in our study is between the long-term control and the stop-selection. However, because they were conducted at different times there may have been unknown differences in lab conditions, which is why we started the short-term control at the same time as the stop selection treatment. The short-term control produced similar results compared to the long-term control. We have now explained this in the manuscript (Introduction section and subsection “’Stop-selection’ experiment”).

Results: The results jump among experiments and it is unclear which experiments are being discussed. Maybe it is best to explain what happened in the Control Experiment (all results). Then compare it to the other experiments.

As suggested, we have now modified both the results and the figures and presented all the results from each experiment in the following sequence: long-term control, stop-selection and short-term control.

At all the times the manuscript should refer to the experiments by their established names, e.g subsection “Long-term control” should say the aseasonal experiment. Why is this result presented before the no-selection experiment? Before Figure 2?

As explained above, we have now moved the aseasonal experiment to the appendix.

*In the same paragraph there are experiments that were never introduced*

Sorry for the mistake. We have now removed this experiment from the manuscript.

Paragraph two? For which experiment? the long or short term?

For the long-term control above, as indicated by the subheading. We believe this is not an issue anymore because now all the results from each experiment are presented separately under different subheadings.

Subsection “Short-term control”. I don't think initial population size of the experiments is mentioned anywhere except here in the results. Why present a complication concerning the stop-selection experiment before presenting the stop-selection experiment or their results?

As suggested by the reviewer, we have explained the differences in initial population size and presented the stop-selection before the short-term control.

Subsection “’Stop selection’ experiment following the order of these figures is exhausting and confusing.

We are sorry for the confusion. Now, each figure presents the main results for each experiment, and the figures match the order they are presented in the Introduction, Results and Materials and methods.

Figures:

Titles don't match content (see Figure 3 and Figure 6 for example).

We have now fixed this in the manuscript.

Consider a large figure that includes all experiments (combines Figure 2 and Figure 3 and Figure 7). If the 4 other experiments are all compared to the long-term control then they should all be in the same figure.

As indicated before, each figure presents all results for each experiment, but we have moved the aseasonal experiment to the appendix.

*Discussion*

The Discussion is only big picture and skips any discussion of many of the results (no mention of the reduced food experiment results at all….) This disconnects the work from the bigger interpretation and feels unbalanced.

As explained above, we have removed the reduced-food experiment from the manuscript. However, we have now discussed some of our results in more detail (Results section).

Discussion paragraph two, I am not sure what 'at ecological scales' means Do you mean timescales? If so there are numerous reviews with hundreds of examples of rapid evolution of ecologically relevant traits (Hendry, Kinison, Reznick, Thompson…). Whether these have been robustly shown to impact ecological dynamics is less clear but see good experimental examples by Willliams Levine 2016, Turcotte et al. 2011 or 2103, Becks multiple, Yoshida et al. 2003 and numerous others by Hairston).

What we meant is that there are few studies showing rapid evolutionary changes that, at the same time, feedback to affect population dynamics. We have now clarified this in the manuscript (Discussion section, paragraph one).

A lot of the Discussion concerns delayed density dependence but this was barely mentioned in the Introduction.

As suggested, we have now explained why delayed density dependence is important in our system (Introduction section).

Immediately when I saw the title of the manuscript, and especially the Abstract, I thought of Paul Schmidt's work (U Penn) on seasonal adaptation in Drosophila. Although the authors cited one of their studies I think the links to this body of work should be more prominent as they have discussed evolutionary changes over multiple generations in a lab experiment that are driven by life-history trade-offs (Schmidt Conde 2006, Evolution).

Thank you for the suggestion. We have now included and discussed this and other studies from the Schmidt’s group in the manuscript (Discussion section).

[Editors' note: further revisions were requested prior to acceptance, as described below.]

The manuscript has been improved but there are some remaining issues that need to be addressed before acceptance, as outlined below:

1) The revised manuscript includes comments and changes in response to reviewer 3, but the comments by reviewer 1 and 2 have not been taken into consideration when revising the manuscript (although commented on when appealing the first decision). The authors should make those changes as well.

We have now made the changes requested by reviewers 1 and 2 as well as the changes we suggested to the editor in our appeal letter.

2) In addition, the revised version needs further restructuring and clarification as it is still not clear from reading what experiments have been done and why (e.g., add an overview figure).

We have now added an overview figure (now Figure 1) of all three experiments to highlight what experiments have been done and their purpose.

We also want to note that, as requested by reviewer 3, we have made extensive changes in the manuscript to improve clarity. Specifically, we have added a new paragraph in the Introduction to describe the experiments and link them with our predictions. For example, we wrote: “In the second experiment, we tested the role of viability selection during the nonbreeding season by tracking 13 additional populations over 31 generations using a similar protocol to the ‘long-term control’…(the ‘stop-selection’ treatment).” and wrote: “In order to address potential environmental changes in the lab, we conducted a third experiment(…) This 'short-term control' experiment also had 13 replicate populations tracked over 31 generations.”

We have also made changes in the results and analysis to remind the reader about the goal of each experiment. For example, in the results, we wrote: “To investigate the presence of viability selection for small body size and whether this selection was density-dependent, we measured female dry weight in 38 generations from 25 different populations.”

3) Also, in the paragraph 'Delayed density dependence' of the Results section, the authors compare the results of independent correlations between selection differentials and population sizes. A formal statistical test such as ANCOVA or mixed effect model is need for this comparison and the authors have all the data.

We believe there is a misunderstanding here. The delayed density dependence analysis only used population size as a response variable. We have clarified this point in the text (subsection “Delayed density dependence”). The editor might be referring to the analysis in the Appendix 3—table 1 and Figure 9, where we used two linear mixed effect modes to compare the effects of density on body size before and after the non-breeding season. As suggested, we have now used a single linear mixed effect model with body size as response variable and the interaction between density and whether body size was measured before or after the non-breeding season as fixed effect (Subsection “Long-term control”). Generation and population were entered as random effects. The results are consistent with what we presented before: changes in body size were driven by changes in mean dry weight after the non-breeding season rather than changes in the mean dry weight before the non-breeding season.